# PARP10 promotes the repair of nascent strand DNA gaps through RAD18 mediated translesion synthesis

Jude B. Khatib[1,2], Ashna Dhoonmoon [1,2], George-Lucian Moldovan [1] & Claudia M. Nicolae [1] ✉

Replication stress compromises genomic integrity. Fork blocking lesions such as those induced by cisplatin and other chemotherapeutic agents arrest replication forks. Repriming downstream of these lesions represents an important mechanism of replication restart, however the single stranded DNA (ssDNA) gaps left behind, unless efficiently filled, can serve as entry point for nucleases. Nascent strand gaps can be repaired by BRCA-mediated homology repair. Alternatively, gaps can also be filled by translesion synthesis (TLS) polymerases. How these events are regulated is still not clear. Here, we show that PARP10, a poorly-characterized mono-ADP-ribosyltransferase, is recruited to nascent strand gaps to promote their repair. PARP10 interacts with the ubiquitin ligase RAD18 and recruits it to these structures, resulting in the ubiquitination of the replication factor PCNA. PCNA ubiquitination, in turn, recruits the TLS polymerase REV1 for gap filling. We show that PARP10 recruitment to gaps and the subsequent REV1-mediated gap filling requires both the catalytic activity of PARP10, and its ability to interact with PCNA. We moreover show that PARP10 is hyperactive in BRCA-deficient cells, and its inactivation potentiates gap accumulations and cytotoxicity in these cells. Our work uncovers PARP10 as a regulator of ssDNA gap filling, which promotes genomic stability in BRCA-deficient cells.

Preservation of genome integrity is an essential component of tumor suppression. Mutations in DNA repair genes, such as BRCA1 and BRCA2, result in genomic instability and increased cancer susceptibility. On the other hand, many chemotherapeutic approaches employ genotoxic agents such as cisplatin to cause irremediable DNA damage in cancer cells.

A subset of these agents, including cisplatin, cause the formation of DNA adducts, thereby interfering with normal DNA replication. Ongoing replication forks can arrest upon encountering these DNA lesions. Since prolonged fork arrest can result in fork breakage and double strand break (DSB) formation[1,2], cells have adopted mechanisms to stabilize and restart the arrested forks. One way this can be achieved is through fork reversal, a process in which the two nascent strands anneal to each other[3], stabilizing the fork and eventually allowing replication restart using the nascent strand of the sister chromatid as template. Another fork restart mechanism involves repriming downstream of the lesion, catalyzed by the primase-polymerase PRIMPOL, leaving behind a single stranded DNA (ssDNA) gap in the nascent strand, to be filled at a later time[4–15].

Nascent strand ssDNA gaps have received renewed interest in recent years, since their accumulation was shown to correlate with the sensitivity of cancer cells, particularly those with BRCA mutations, to genotoxic chemotherapeutic agents such as cisplatin or PARP inhibitors[5,8,16–28]. The process through which ssDNA gaps become

[1]Department of Biochemistry and Molecular Biology, The Pennsylvania State University College of Medicine, Hershey, PA 17033, USA. [2]These authors contributed equally: Jude B. Khatib, Ashna Dhoonmoon. ✉e-mail: cmn14@psu.edu

cytotoxic is still unclear. Nevertheless, recent studies have identified several mechanisms of gap filling, which may suppress this cytotoxicity[5,8,16–32]. One involves homology-dependent gap filling, using the nascent strand of the sister chromatid, through either template switching or BRCA-mediated homologous recombination repair. The other one entails gap filling by specialized translesion synthesis (TLS) polymerases, such as the REV1- Polζ complex. Importantly, it was shown that BRCA-deficient cells rely on TLS-mediated gap filling[7,8]. How this regulation is achieved remains unclear.

Translesion synthesis was initially identified as a mutagenic mechanism bypassing DNA lesions. TLS polymerases can synthesize across certain DNA lesions, albeit in a potentially mutagenic manner, thereby allowing replication to continue without interruption at the lesion[33,34]. TLS is initiated by the ubiquitination of PCNA, a homotrimeric ring-shaped factor that encircles DNA and provides processivity to replication polymerases[35–37]. At stalled replication forks, mono-ubiquitination of PCNA induces a switch from replicative polymerases to TLS polymerases, some of which contain separate domains for interaction with PCNA (known as PIP-box motifs) and ubiquitin. RAD18 is the major E3 ubiquitin ligase for PCNA ubiquitination, while the ubiquitin-specific protease USP1 was shown to promote its de-ubiquitination[38,39]. In addition to lesion bypass, recent studies showed that RAD18-mediated PCNA ubiquitination and the REV1 TLS polymerase also promote filling of ssDNA gaps during G2 phase[7,8].

We previously identified the mono-ADP-ribosyltransferase PARP10 as a regulatory component of PCNA-mediated TLS[40–42]. We showed that PARP10 contains a PIP-box motif through which it interacts with PCNA. PARP10 depletion reduced the amount of PCNA ubiquitination upon exposure to a high dose of hydroxyurea (HU) which arrests replication forks. Moreover, PARP10 depletion also reduced the mutagenic bypass of UV-induced DNA lesions on a plasmid substrate transfected into cells. These findings suggested that PARP10 promotes mutagenic lesion bypass by enhancing PCNA ubiquitination, although the mechanism underlying this effect remained unclear.

PARP10 belongs to a subset of ADP-ribosyltransferases which, unlike PARP1, can only catalyze the transfer of a single ADP-ribose molecule on substrates, a process termed mono-ADP-ribosylation (MARylation)[43–47]. PARP10 was originally identified as a Myc-interacting protein[48], and found to be involved in varied processes including G1/S cell cycle transition[49], caspase-dependent apoptosis[50], cytokine-induced activation of NFκB pathway[51], mitochondrial oxidation[52] and cell migration[53]. In general, the underlying mechanisms and relevant ADP-ribosylation substrates responsible for these functions are not known. Importantly, we previously showed that PARP10 is overexpressed in 20-30% of breast and ovarian tumors, suggesting a role in cellular transformation in those tissues[41].

Here, we show that PARP10 is an important component of the ssDNA gap filling process in human cells. We find that PARP10 deficiency results in the accumulation of replication stress-induced ssDNA gaps. PARP10 localizes to ssDNA gaps through its PCNA-interacting motif, in a process that also requires its catalytic activity. We next show that PARP10 interacts with and mono-ADP-ribosylates RAD18, and recruits it to these structures, to promote PCNA ubiquitination. PARP10 inactivation reduces PCNA ubiquitination at ssDNA gaps, impairing the recruitment of REV1. Moreover, we show that BRCA-deficient cells rely on PARP10-mediated gap filling. Finally, we report that concomitant inactivation of PARP10 and the BRCA pathway results in ssDNA gap accumulation and reduced cellular survival.

## Results

### PARP10 is required for suppressing the accumulation of ssDNA gaps in BRCA-proficient cells

Since our previous work indicated a role for PARP10 in promoting PCNA ubiquitination-mediated mutagenesis[40,41], we sought to investigate the impact of PARP10's loss on ssDNA gap formation during DNA replication. In both HeLa and DLD1 cells, PARP10 depletion resulted in increased ssDNA gap formation (Fig. 1a–c), as measured using the BrdU alkaline comet assay, previously used by us and others to measure replication-associated gaps[23,54,55]. Gap induction upon depletion of PARP10 was observed under two different experimental conditions previously shown to induce ssDNA gaps[8,18,23,25,56,57], namely treatment with low-dose (0.4 mM) hydroxyurea (HU) or exposure to 150 μM cisplatin. BRCA2-knockout cells, which we previously showed to accumulate ssDNA gaps[25] were employed as positive control for detecting gap accumulation under these conditions. Two different siRNA oligonucleotides were used to deplete PARP10. While western blots indicated that both oligonucleotides reduce PARP10 protein levels, quantitative reverse transcription PCR (RT-qPCR) analyses of mRNA levels showed that siPARP10[#2] was able to deplete PARP10 mRNA levels to a higher degree than siPARP10[#1] (Supplementary Fig. 1a, b). This difference is reflected in the stronger impact of siPARP10[#2] on gap formation (Fig. 1a, c).

To confirm that PARP10-depleted cells accumulate nascent strand discontinuities, we also employed the S1 nuclease DNA fiber combing assay, which can specifically detect ssDNA gaps in the nascent strand[58]. Unlike control cells, PARP10 depleted cells showed a reduction in the CldU/IdU ratio in the S1 nuclease-treated samples, indicating accumulation of nascent strand ssDNA gaps, upon exposure to either 0.4 mM HU (Fig. 1d–g), or 150 μM cisplatin (Fig. 1h). BRCA2-knockout cells were employed as a positive control for detecting nascent strand gap accumulation under these conditions. To confirm these results which were obtained using siRNA-mediated PARP10 knockdown, we also generated PARP10-knockout HeLa cells using CRISPR/Cas9-mediated genome editing (Supplementary Fig. 1b). Similar to PARP10 depletion, PARP10-knockout cells showed nascent strand ssDNA gap formation upon HU (Fig. 1f, g) or cisplatin (Fig. 1i) treatment. Depletion of fork reversal translocases ZRANB3 and SMARCAL1 did not affect HU-induced ssDNA gap accumulation in PARP10-knockout cells (Supplementary Fig. 2), arguing that the gaps do not occur on reversed forks.

Recently, a PARP10 specific inhibitor, namely OUL35, has been developed and became commercially available[59]. We thus sought to employ a pharmacological approach in order to validate the results described above obtained with genetic depletion of PARP10. Treatment of HeLa cells with OUL35 resulted in HU-induced ssDNA gap formation (Fig. 1j), implying that the catalytic activity of PARP10 is required for its role in gap suppression. Overall, these results indicate that PARP10 plays an important role in suppressing ssDNA gap accumulation induced by genotoxic agents in wildtype cells.

Previous studies have shown that, in BRCA-deficient cells, ssDNA gaps are extended by the MRE11 exonuclease resulting in gap expansion[8,25,60]. In line with this, inhibition of MRE11 exonuclease activity using the specific inhibitor mirin suppressed gap formation in BRCA2-knockout cells, as measured by the S1 nuclease DNA fiber combing assay (Fig. 1k). Similar to the situation in BRCA2-knockout cells, mirin treatment suppressed gap accumulation in both PARP10-depleted and PARP10-knockout cells (Fig. 1k). We recently developed an experimental approach to measure the engagement of MRE11 for gap expansion on nascent DNA, employing the proximity ligation (PLA)-based SIRF (in situ quantification of proteins interactions at DNA replication forks) assay, and showed that MRE11 is specifically recruited to these structures in BRCA-deficient cells[60]. PARP10 depletion also resulted in specific recruitment of MRE11 to nascent DNA under gap-inducing conditions (Fig. 1l). These findings indicate that ssDNA gaps formed in PARP10-deficient cells are processed for expansion by the MRE11 exonuclease, similar to the gaps formed in BRCA-deficient cells.

 

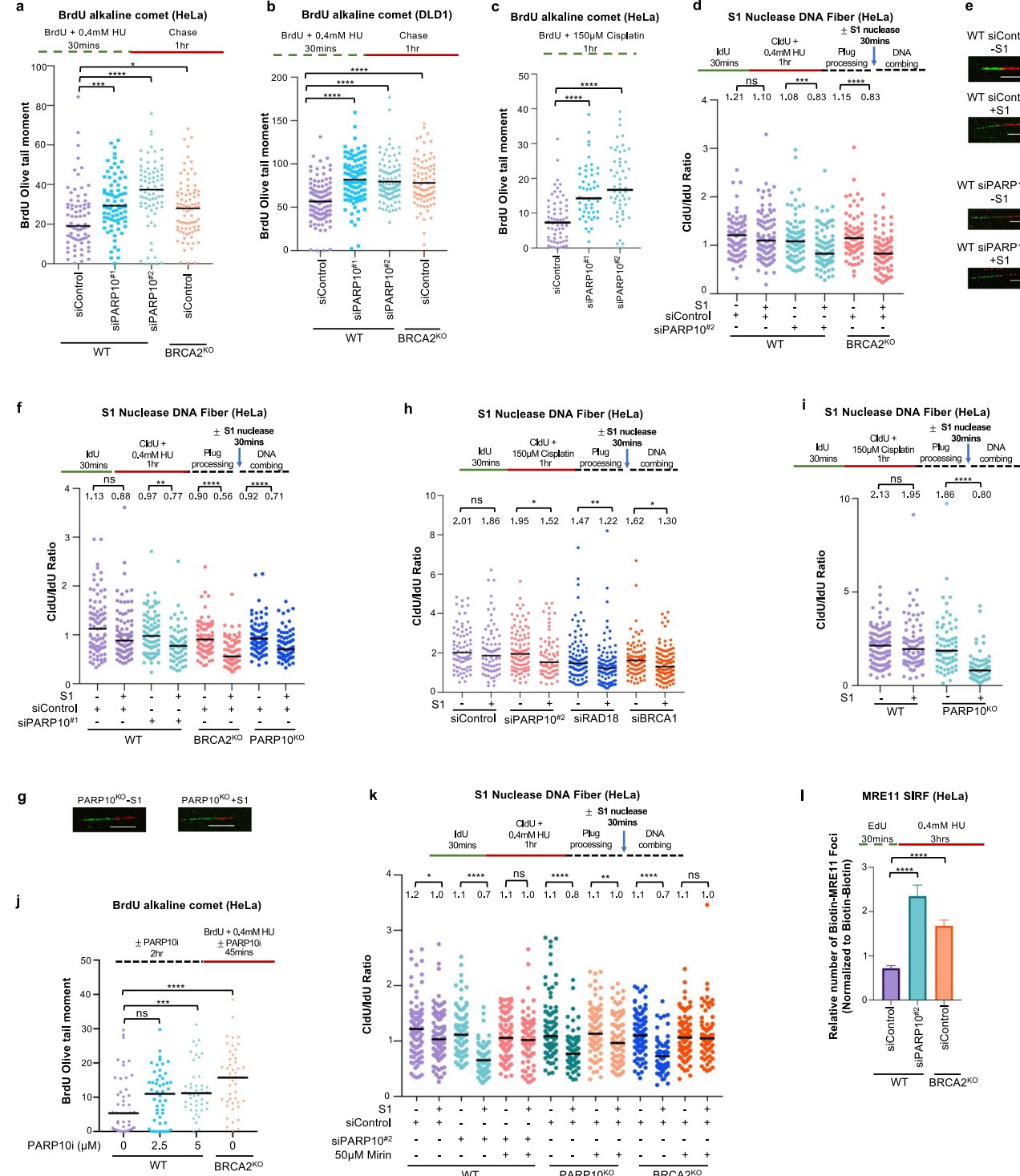

## PARP10 promotes RAD18-mediated PCNA ubiquitination and subsequent REV1 recruitment for gap filling

Since PARP10 depletion caused accumulation of nascent strand gaps, we next sought to investigate if PARP10 may play a direct role in gap filling. We first investigated if PARP10 is itself localized to replication-associated ssDNA gaps. To this end, we treated cells with 150 μM cisplatin to induce ssDNA gaps and labeled nascent DNA with EdU. SIRF assays using a specific PARP10 antibody showed a significant increase in foci formation in cisplatin-treated compared to untreated control cells (Fig. 2a), indicating that PARP10 is recruited to nascent DNA under gap-inducing conditions. PARP10 depletion reduced the number of SIRF foci in cisplatin-treated cells, confirming the specificity of

the SIRF signal. Treatment with 0.4 mM HU, another condition previously shown to induce ssDNA gaps, also resulted in PARP10 binding to nascent DNA in SIRF assays (Fig. 2b, c; Supplementary Fig. 3a). Interestingly, PARP10 inhibition by OUL35 also reduced the PARP10 SIRF signal under these conditions, suggesting that PARP10 catalytic activity promotes its localization to gaps. Overall, these findings indicate that PARP10 binds to nascent strand ssDNA gaps.

Previous studies showed that loss of RAD18, the main ubiquitin ligase responsible for PCNA ubiquitination, causes an increase in ssDNA gaps[7]. In line with this, RAD18-knockdown cells showed ssDNA gap accumulation upon treatment with cisplatin (Fig. 1h). Since depletion of RAD18 or PARP10 resulted in similar phenotypes, we

**Fig. 1 | Loss of PARP10 causes MRE11-mediated accumulation of ssDNA gaps.**
**a**–**c** BrdU alkaline comet assays showing that PARP10 knockdown in HeLa (**a**, **c**) and
DLD1 (**b**) cells causes accumulation of replication-associated ssDNA gaps upon
treatment with 0.4 mM HU (**a**, **b**) or 150 μM cisplatin (**c**), similar to BRCA2 deletion.
At least 50 nuclei were quantified for each condition. The median values are marked
on the graph and listed at the top. Asterisks indicate statistical significance (Mann-
Whitney, two-tailed). Schematic representations of the assay conditions are shown
at the top. Western blots and RT-qPCR confirming PARP10 knockdown are shown in
Supplementary Fig. 1a, b. **d**–**i** S1 nuclease DNA fiber combing assays showing that
PARP10 knockdown (**d**, **e**, **f**, **h**) or knockout (**f**, **g**, **i**) in HeLa cells causes accumu-
lation of nascent strand ssDNA gaps upon treatment with 0.4 mM HU (**d**–**g**) or
150 μM cisplatin (**h**, **i**) similar to BRCA deficiency. Quantifications (**d**, **f**, **h**, **i**) and
representative micrographs, with scale bars representing 10 μm (**e**, **g**) are shown.
The ratio of CldU to IdU tract lengths is presented, with the median values marked
on the graphs and listed at the top. At least 63 tracts were quantified for each
sample. Asterisks indicate statistical significance (Mann-Whitney, two-tailed).
Schematic representations of the assay conditions are shown at the top. Western
blots confirming PARP10 knockout are shown in Supplementary Fig. 1b. **j** BrdU
alkaline comet assay showing that PARP10 inhibition using the specific inhibitor
OUL35 causes accumulation of replication-associated ssDNA gaps upon treatment
with 0.4 mM HU in HeLa cells. At least 44 nuclei were quantified for each condition.
The median values are marked on the graph and listed at the top. Asterisks indicate
statistical significance (Mann-Whitney, two-tailed). A schematic representation of
the assay conditions is shown at the top. **k** S1 nuclease DNA fiber combing assays
showing that inhibition of MRE11 endonuclease activity using the specific inhibitor
mirin suppresses the accumulation of ssDNA gaps induced by treatment with
0.4 mM HU in PARP10-deficient cells. The ratio of CldU to IdU tract lengths is
presented, with the median values marked on the graphs and listed at the top.
At least 65 tracts were quantified for each sample. Asterisks indicate statistical
significance (Mann-Whitney, two-tailed). A schematic representation of the assay
conditions is shown at the top. **l** SIRF experiment showing that treatment with
0.4 mM HU induces binding of MRE11 to nascent DNA in PARP10-depleted HeLa
cells, similar to BRCA-knockout cells. At least 92 cells were quantified for each
condition. Bars indicate the mean values, error bars represent standard errors of
the mean, and asterisks indicate statistical significance (*t*-test, two-tailed, unpaired).
A schematic representation of the assay conditions is shown at the top. Source data
are provided as a Source Data file.

sought to investigate if these two factors act together in gap suppres-
sion. We thus investigated if PARP10 and RAD18 interact with each
other. Co-immunoprecipitation experiments showed that PARP10 spe-
cifically co-precipitates with RAD18 (Fig. 2d). Treatment with 0.4 mM
HU or 150 μM cisplatin resulted in only a very minor, if any, increase in
the PARP10-RAD18 interaction. Since co-immunoprecipitation experi-
ments are qualitative assays, we next investigated the PARP10-RAD18
interaction using a quantitative approach, namely the PLA assay. We
observed a specific signal for the PARP10-RAD18 interaction in this assay
as well. Treatment with 0.4 mM HU resulted in an increase in the PLA
signal (Fig. 2e-g; Supplementary Fig. 3b), suggesting that PARP10 and
RAD18 interact in response to ssDNA gap induction. Depletion of RAD18
and of PARP10 reduced the number of PLA foci, confirming the speci-
ficity of the PLA signal.

Since our studies described above indicated that PARP10 is
recruited to ssDNA gaps and interacts with RAD18 under these con-
ditions, we next investigated the impact of PARP10 on RAD18 locali-
zation to gaps. RAD18 SIRF assays showed that treatment with 150 μM
cisplatin increased the number of foci (Fig. 2h), indicating that RAD18
binds to nascent DNA gaps. Depletion of PARP10 reduced the
RAD18 signal. Similar findings were obtained upon treatment with
0.4 mM HU (Fig. 2i, j). Overall, these findings suggest that PARP10
promotes the recruitment of RAD18 to nascent strand gaps. Interest-
ingly, RAD18 depletion, in turn, increased PARP10 SIRF signal upon
treatment with 0.4 mM HU (Fig. 2b). This potentially reflects the
accumulation of ssDNA gaps in RAD18-deficient cells, caused by
defective PCNA ubiquitination-mediated gap filling.

RAD18 is the main ubiquitin ligase for PCNA. Since our results
described above indicated that RAD18 is recruited to nascent strand
gaps, we reasoned that this recruitment of RAD18 may result in ubi-
quitination of PCNA at ssDNA gaps. To test this, we employed the SIRF
assay to measure PCNA ubiquitination at ssDNA gaps. Using an anti-
body specifically detecting the ubiquitinated form of PCNA, we were
able to observe SIRF foci upon treatment of HeLa cells with 0.4 mM HU
(Fig. 3a). Depletion of the PCNA ubiquitin ligase RAD18 reduced the
number of ubiquitinated PCNA SIRF foci, while depletion of the PCNA
deubiquitinating enzyme USP1 increased these foci (Fig. 3b; Supple-
mentary Fig. 1c, d; Supplementary Fig. 3c). These controls confirmed
the specificity of the ubiquitinated PCNA SIRF signal. Importantly,
PARP10 depletion, deletion or inhibition resulted in a decrease in
ubiquitinated PCNA SIRF foci (Fig. 3a–c), arguing that PARP10 pro-
motes the ubiquitination of PCNA at nascent strand gaps.

PCNA ubiquitination promotes the recruitment of TLS poly-
merases to bypass DNA lesions[35–37]. Recently, the TLS polymerase REV1
was shown to participate in gap filling[7,8]. We thus investigated if the
reduction in RAD18 recruitment and subsequent PCNA ubiquitination
upon PARP10 loss translates into reduced REV1 engagement on nas-
cent strand gaps. REV1 SIRF experiments showed that REV1 binds
nascent DNA under gap inducing conditions (0.4 mM HU or 150 μM
cisplatin), and this binding is reduced upon PARP10 depletion
(Fig. 3d–f). These findings indicate that, by enhancing RAD18 recruit-
ment to nascent DNA, PARP10 promotes PCNA ubiquitination and
subsequently the recruitment of REV1 for gap filling.

To further test this model, we investigated if PARP10 and RAD18
are epistatic for ssDNA gap suppression. We created RAD18-knockout
HeLa cells by CRIPSR/Cas9 gene editing (Supplementary Fig. 1e). As
expected, these cells showed increased ssDNA gaps upon treatment
with 150 μM cisplatin or 0.4 mM HU (Fig. 3g, h). Depletion of PARP10 in
HeLa-wildtype cells caused increased gap accumulation. In contrast,
depletion of PARP10 in HeLa-RAD18^KO cells did not further increase the
amount of ssDNA gaps, indicating that PARP10 and RAD18 are epistatic
for gap suppression. As control, BRCA2 depletion, as expected, caused
increased gap accumulation in both wildtype and RAD18-knockout
cells. Overall, these findings argue that PARP10 promotes gap sup-
pression through RAD18-mediated PCNA ubiquitination.

### The catalytic activity of PARP10 and its ability to interact with PCNA are both required for PARP10 localization to ssDNA gaps and for its role in gap suppression

PARP10 has mono-ADP-ribosyltransferase catalytic domain in its
C-terminus (Fig. 4a). We previously identified a PCNA-interacting PIP-
box motif next to this domain and showed that inactivation of either
the catalytic activity or of the PCNA interaction reduced UV-induced
mutagenesis as measured using the SupF shuttle vector assay[40,41]. To
investigate the relevance of the PCNA interaction and of the catalytic
activity of PARP10 in ssDNA gap metabolism, we stably re-expressed
wildtype, catalytic site mutant (G888W), and ΔPIP (V834A, F837A,
Y838A triple mutant altering the PIP box sequence from QEVVRAFY to
QEVARAAA) PARP10 variants in PARP10^KO HeLa cells under the control
of the SV40 promoter using a lentiviral system. The exogenous
wildtype and mutant PARP10 variants were expressed at similar
levels, albeit these levels were higher than that of endogenous
PARP10 (Fig. 4b).

We first investigated the impact of these PARP10 mutations on
its recruitment to ssDNA gaps. Compared to wildtype PARP10, the
G888W catalytic site mutant and the PIP-box mutant showed a sig-
nificant reduction in PARP10 localization to nascent DNA under
ssDNA gap inducing conditions, including both 0.4 mM HU as well
as 150 μM cisplatin treatment. (Fig. 4c, d). We previously showed
that PARP10 interacts with ubiquitinated PCNA using its UIM

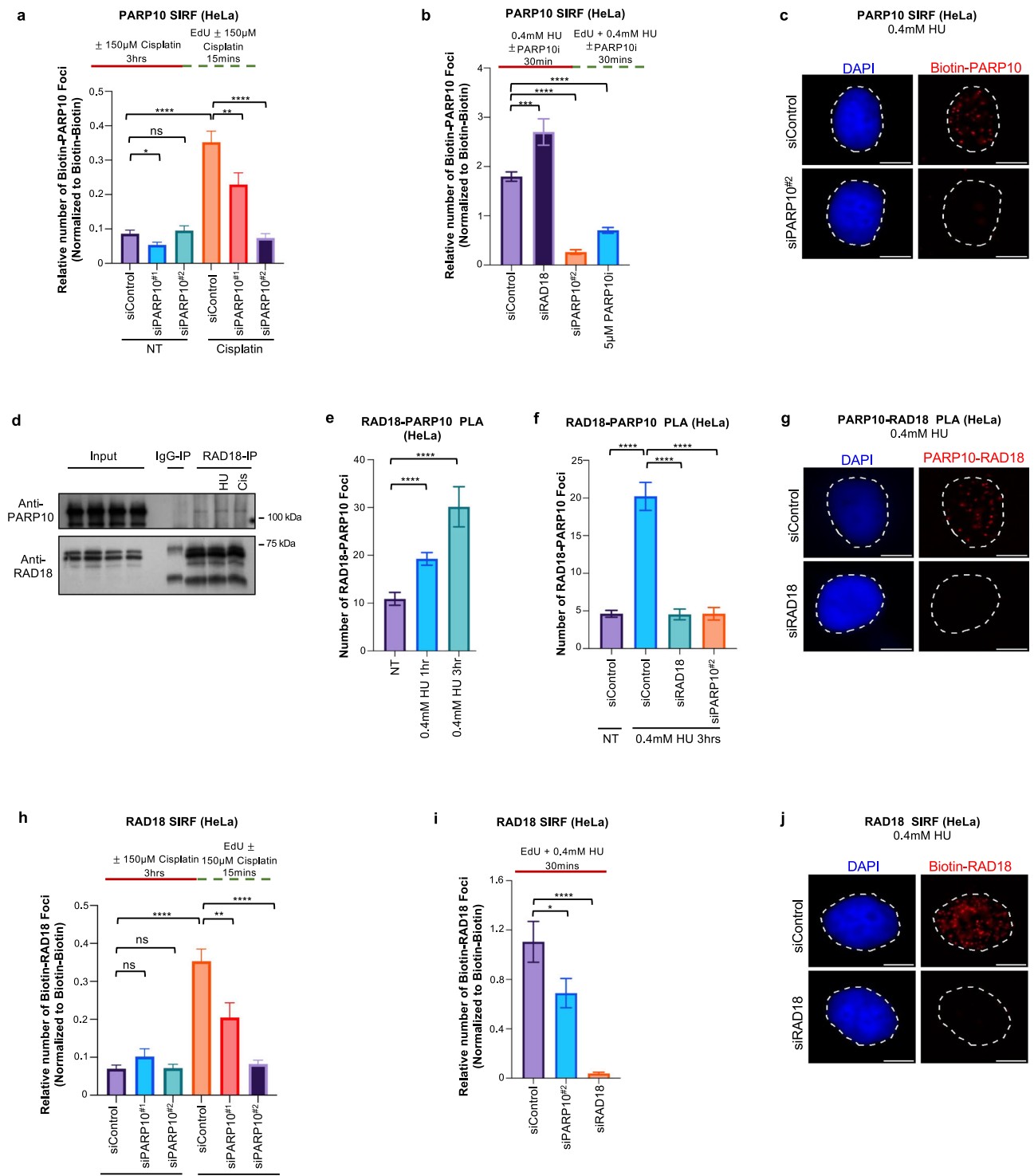

**Fig. 2 | PARP10 binds to nascent strand gaps and promotes the recruitment of RAD18 to these structures. a–c** SIRF experiments showing that treatment with 150 μM cisplatin (**a**) or 0.4 mM HU (**b, c**) induces binding of PARP10 to nascent DNA in HeLa cells. Quantifications (**a, b**) and representative micrographs, with scale bars representing 10 μm (**c**) are shown. At least 76 cells were quantified for each condition. Bars indicate the mean values, error bars represent standard errors of the mean, and asterisks indicate statistical significance (*t*-test, two-tailed, unpaired). Schematic representations of the assay conditions are shown at the top. **d** Co-immunoprecipitation experiment in HeLa cells showing that PARP10 co-precipitates with RAD18. Cells were treated with 0.4 mM HU, 150 μM cisplatin, or left untreated as indicated. **e–g** PLA assays showing that RAD18 and PARP10 co-localize upon treatment with 0.4 mM HU in HeLa cells. Knockdown of PARP10 or RAD18 is used as control to confirm the specificity of the PLA signals observed.

Quantifications (**e, f**) and representative micrographs, with scale bars representing 10 μm (**g**) are shown. At least 45 cells were quantified for each condition. Bars indicate the mean values, error bars represent standard errors of the mean, and asterisks indicate statistical significance (*t*-test, two-tailed, unpaired). **h–j** SIRF experiments showing that PARP10 depletion reduces the binding of RAD18 to nascent DNA upon treatment with 150 μM cisplatin (**h**) or 0.4 mM HU (**i, j**) in HeLa cells. Quantifications (**h, i**) and representative micrographs, with scale bars representing 10 μm (**j**) are shown. At least 70 cells were quantified for each condition. Bars indicate the mean values, error bars represent standard errors of the mean, and asterisks indicate statistical significance (*t*-test, two-tailed, unpaired). Schematic representations of the assay conditions are shown at the top. Source data are provided as a Source Data file.

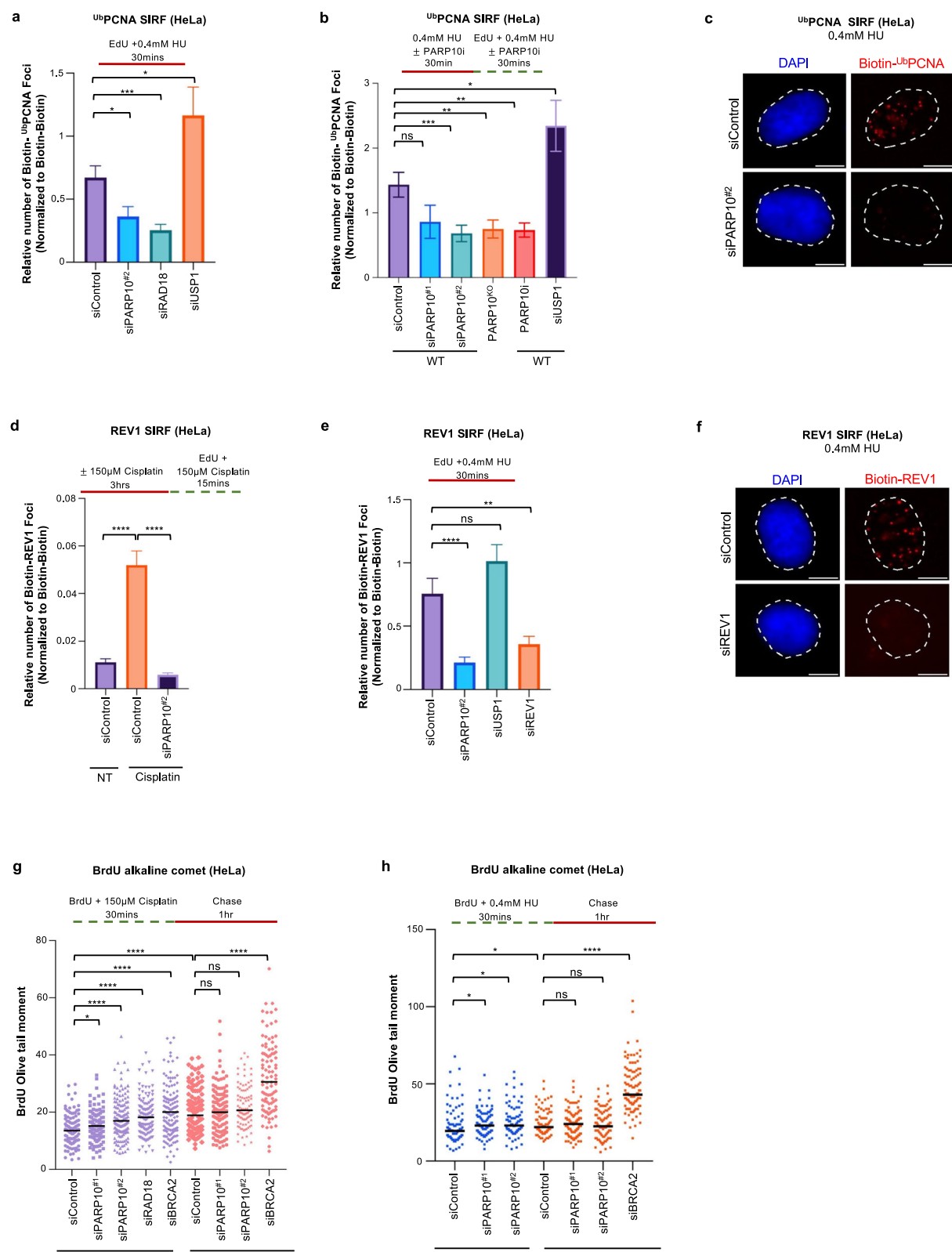

domains[40]. To test the impact of this interaction of PARP10 localization, we also re-expressed in PARP10[KO] HeLa cells a PARP10 variant with deletion of UIM domains (aminoacids 650–690) (Supplementary Fig. 4a). SIRF experiments indicated that this mutant is not defective in PARP10 recruitment to nascent DNA upon treatment with 0.4 mM HU (Supplementary Fig. 4b–d). Overall, these findings indicate that both the catalytic activity and the PCNA

interaction, but not the ubiquitin interaction, are required for PARP10 recruitment to nascent strand ssDNA gaps.

We next investigated the impact of the PARP10 mutants on ssDNA gaps. BrdU alkaline comet assays indicated that, both in response to 0.4 mM HU as well as 150 μM cisplatin treatment, stable re-expression of wildtype PARP10 suppressed ssDNA gap accumulation in PARP10[KO] cells. In contrast, stable expression of either the G888W catalytic site

**Fig. 3 | Loss of PARP10 suppresses PCNA ubiquitination and REV1 recruitment to nascent strand gaps. a–c** SIRF experiments showing that PARP10 depletion, deletion or inhibition reduces the levels of ubiquitinated PCNA at nascent DNA gaps induced by treatment with 0.4 mM in HeLa cells. Quantifications (**a**, **b**) and representative micrographs, with scale bars representing 10 μm (**c**) are shown. At least 53 cells were quantified for each condition. Bars indicate the mean values, error bars represent standard errors of the mean, and asterisks indicate statistical significance (*t*-test, two-tailed, unpaired). Schematic representations of the assay conditions are shown at the top. Western blots confirming knockdowns of RAD18 and USP1, which are used as controls, are shown in Supplementary Fig. 1c, d. **d–f** SIRF experiments showing that PARP10 depletion reduces the binding of REV1 to nascent DNA upon treatment with 150 μM cisplatin (**d**) or 0.4 mM HU (**e**, **f**) in HeLa cells. Quantifications (**d**, **e**) and representative micrographs, with scale bars representing 10 μm (**f**)

are shown. At least 63 cells were quantified for each condition. Bars indicate the mean values, error bars represent standard errors of the mean, and asterisks indicate statistical significance (*t*-test, two-tailed, unpaired). Schematic representations of the assay conditions are shown at the top. **g**, **h**. BrdU alkaline comet assays showing that PARP10 depletion causes accumulation of replication-associated ssDNA gaps upon treatment with 150 μM cisplatin (**g**) or 0.4 mM HU (**h**) in wildtype, but not in RAD18-knockout HeLa cells. At least 73 nuclei were quantified for each condition. The median values are marked on the graph and listed at the top. Asterisks indicate statistical significance (Mann-Whitney, two-tailed). Schematic representations of the assay conditions are shown at the top. Western blots confirming RAD18 knockout are shown in Supplementary Fig. 1e. Source data are provided as a Source Data file.

mutant or the PIP-box mutant failed to suppress ssDNA gap accumulation in PARP10$^{KO}$ cells (Fig. 4e,f). We also investigated gap formation using the S1 nuclease DNA fiber combing assay. Similar to the BrdU alkaline assay results, exogenous re-expression of wildtype PARP10 could reduce ssDNA gap accumulation in PARP10$^{KO}$ HeLa cells upon 0.4 mM HU treatment, but the catalytic site mutant and the PIP-box mutant did not (Fig. 4g, h). These findings indicate that both the PCNA interaction and the catalytic activity of PARP10 are required for ssDNA gap suppression.

Since, as described above, we found that PARP10 interacts with RAD18 and PARP10-deficient cells showed reduced RAD18 recruitment to nascent DNA under ssDNA gap-inducing conditions, we next measured its localization in the PARP10 mutants. Upon exposure to 0.4 mM HU, both the G888W catalytic site mutant and the PIP-box mutant showed reduced RAD18 recruitment to nascent DNA compared to wildtype controls, (Fig. 4i). We then investigated if these mutants have deficient interaction with RAD18. Co-immunoprecipitation experiments suggested that the PIP-box mutant may have reduced interaction with RAD18 (Fig. 4; Supplementary Fig. 5a, b). To further evaluate this, we employed quantitative PLA experiments. Re-expression of wildtype PARP10 in PARP10-knockout cells restored the RAD18-PARP10 PLA signal (Supplementary Fig. 6). We next investigated the PARP10 mutants. Compared to wildtype PARP10, the PIP-box mutant was defective in RAD18 interaction, while the G888W catalytic site mutant retained normal RAD18 interaction levels (Fig. 4k; Supplementary Fig. 5c). These findings suggest that PARP10 recruitment to PCNA promotes or stabilizes its interaction with RAD18.

Finally, we investigated the localization of the TLS polymerase REV1 to nascent strand gaps, since REV1 recruitment depends on RAD18-mediated PCNA ubiquitination and the results presented above indicated that PARP10 promotes this recruitment. We found that cells expressing either the G888W catalytic site mutant or the PIP-box mutant showed reduced recruitment of REV1 to nascent DNA under ssDNA gaps-inducing conditions, compared to cells expressing wildtype PARP10 (Fig. 4l). Overall, these findings are in line with the reduced binding of the PARP10 mutants to nascent DNA under these conditions, and further reinforce our findings that PARP10 directly promotes the recruitment of RAD18 to stalled forks and subsequent REV1-mediated gap filling upon PCNA ubiquitination.

## PARP10 mono-ADP-ribosylates RAD18

Since the catalytic activity of PARP10 was required for RAD18 recruitment to nascent DNA, we asked if PARP10 is able to mono-ADP-ribosylate RAD18. We employed a recombinant PARP10 C-terminal fragment (805-1025) encompassing the PARP catalytic domain. We incubated this fragment with full-length recombinant RAD18 in the presence of biotin-labeled NAD$^+$. We monitored the transfer of biotinylated mono-ADP-ribose on RAD18 using streptavidin-HRP blots. We observed that RAD18 is robustly mono-ADP-ribosylated by PARP10 in this system (Fig. 5a; Supplementary Fig. 7a). As expected[44], PARP10 was

able to mono-ADP-ribosylate itself. As controls, no signal was observed in the absence of biotin-NAD$^+$ or of PARP10.

Next, we tested if RAD18 may be mono-ADP-ribosylated by PARP10 in cells. We employed the AbD33204 antibody recently developed to detect MARylated substrates[61]. We detected a specific signal when using these antibodies in conjunction with RAD18 antibodies in PLA assays (Fig. 5b–d). The signal was increased by treatment with 0.4 mM HU for 3hrs and was reduced in PARP10-knockout cells. Expression of wildtype PARP10, but not of the G888W catalytic site mutant or the PIP-box mutant restored the RAD18-MAR signal in PARP10$^{KO}$ cells (Fig. 5e; Supplementary Fig. 7b). While we cannot rule out that the mono-ADP-ribosylation signal detected in these PLA experiments represents MARylation of RAD18-interacting proteins rather than of RAD18 itself, these results, together with the in vitro MARylation of RAD18 by PARP10 shown above, suggest that PARP10 mono-ADP-ribosylates RAD18 in cells in response to gap-inducing conditions.

## PARP10 suppresses ssDNA gap accumulation in BRCA-deficient cells

Previous studies have shown that the BRCA pathway promotes homology-based filling of ssDNA gaps using the nascent strand of the sister chromatid[5,8,16–28]. Indeed, loss of BRCA1 or BRCA2 caused an increase in ssDNA gap accumulation similar to PARP10 inactivation (Fig. 1). Moreover, we noticed that PARP10 recruitment to nascent DNA upon treatment with 0.4 mM HU is increased by knockdown of BRCA1 or BRCA2 (Fig. 6a, Supplementary Fig. 1f, g), suggesting that PARP10-mediated gap filling is enhanced upon BRCA deficiency. In line with this, SIRF experiments also showed that PARP10 depletion in BRCA2-knockout cells increases the recruitment of MRE11 to nascent DNA under these conditions (Fig. 6b), indicating increased gap expansion upon concomitant loss of PARP10 and the BRCA pathway.

To test this, we measured replication associated gaps, using the BrdU alkaline comet assay, in BRCA-deficient cells upon PARP10 knockdown. Both HU and cisplatin treatment increased gap formation in BRCA2-knockout HeLa cells, and this was further exacerbated by PARP10 depletion (Fig. 6c–e). Similar findings were obtained upon PARP10 depletion in BRCA2-knockout DLD1 cells (Fig. 6f). Moreover, PARP10 depletion in BRCA1-mutant breast tumor-derived MDA-MB-436 cells also increased gap formation upon exposure to 0.4 mM HU or 150 μM cisplatin (Fig. 6g, h). We also validated these findings by employing PARP10-knockout cells. Depletion of BRCA2 in HeLa-PARP10$^{KO}$ cells further enhanced cisplatin-induced gap formation (Fig. 6i). We also knocked out PARP10 in SKOV3 ovarian cancer cells (Supplementary Fig. 1h). PARP10-knockout SKOV3 cells showed increased gap formation upon exposure to 0.4 mM HU, which was further exacerbated by the depletion of BRCA1 or BRCA2 (Fig. 6j). Moreover, depletion of RAD51, the main effector of BRCA-mediated homology-based repair, also increased gap accumulation in PARP10-knockout HeLa cells upon exposure to 0.4 mM HU or 150 μM cisplatin

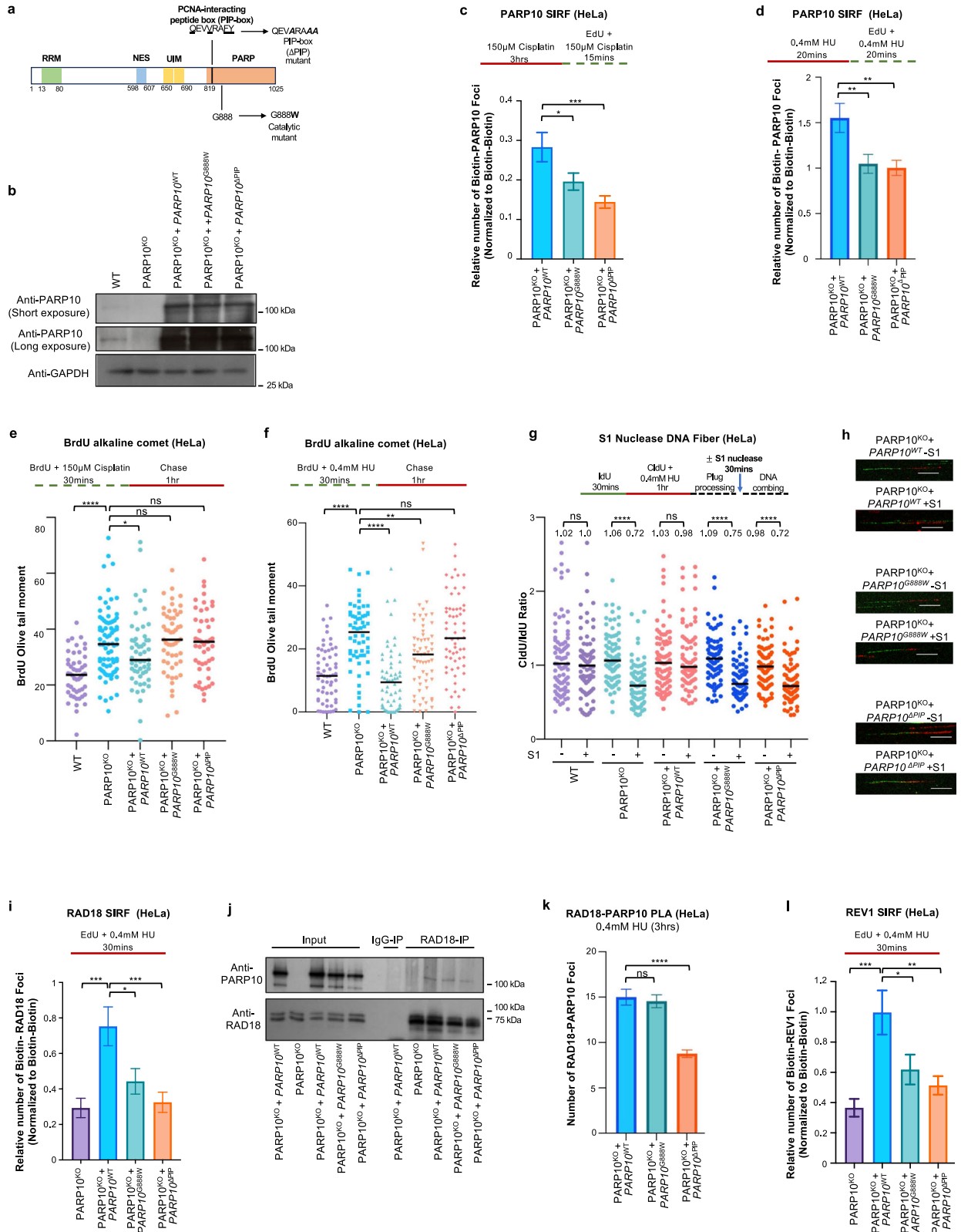

(Supplementary Fig. 8a–c), similar to BRCA2 depletion. Previous studies have shown that loss of FEN1 suppresses HU-induced ssDNA accumulation in BRCA2-deficient cells[18]. While we were able to reproduce these findings, we found that loss of FEN1 did not affect gap accumulation in PARP10-knockout HeLa cells (Supplementary Fig. 9a–c). Altogether, these findings indicate that PARP10 and the BRCA pathway anchor distinct mechanisms of gap suppression, and

simultaneous inactivation of both processes leaves cells severely compromised for gap repair.

Based on these findings, we reasoned that BRCA deficient cells may be hyper-reliant on PARP10-mediated gap filling, thereby providing a window of opportunity for therapeutic intervention in BRCA-mutant cancers. Treatment of BRCA-knockout cells with the PARP10 inhibitor OUL35 further exacerbated HU-induced gap accumulation

**Fig. 4 | PARP10 recruitment to ssDNA gaps and the subsequent TLS-mediated gap filling requires both the PCNA interaction ability and the catalytic activity. a.** Schematic representation of the domain organization of PARP10, indicating the catalytic site mutant (G888W) and the PCNA interaction-deficient (ΔPIP) mutant used. **b.** Western blots showing the exogenous re-expression of wildtype, G888W catalytic site mutant, and ΔPIP PARP10 variants in PARP10-knockout HeLa cells. **c, d** SIRF experiments showing that PARP10 G888W catalytic site mutant and ΔPIP mutant show reduced binding to ssDNA gaps induced by treatment with 150 μM cisplatin (**c**) or 0.4 mM HU (**d**) compared to wildtype control. At least 68 cells were quantified for each condition. Bars indicate the mean values, error bars represent standard errors of the mean, and asterisks indicate statistical significance (*t*-test, two-tailed, unpaired). Schematic representations of the assay conditions are shown at the top. **e, f** BrdU alkaline comet assays showing that re-expression of wildtype PARP10 suppresses the accumulation of replication-associated ssDNA gaps upon treatment with 150 μM cisplatin (**e**) or 0.4 mM HU (**f**) in HeLa PARP10-knockout cells, but re-expression of the G888W catalytic site mutant or of the ΔPIP mutant does not. At least 49 nuclei were quantified for each condition. The median values are marked on the graph and listed at the top. Asterisks indicate statistical significance (Mann-Whitney, two-tailed). Schematic representations of the assay conditions are shown at the top. **g, h** S1 nuclease DNA fiber combing assays showing that re-expression of wildtype PARP10 suppresses the accumulation of nascent strand ssDNA gaps upon treatment with 150 μM cisplatin 0.4 mM HU in HeLa PARP10-knockout cells, but re-expression of the G888W catalytic site mutant or of the ΔPIP mutant does not. Quantifications (**g**) and representative micrographs, with scale bars representing 10 μm (**h**) are shown The ratio of CldU to IdU tract lengths is presented, with the median values marked on the graphs and listed at the top.

At least 70 tracts were quantified for each sample. Asterisks indicate statistical significance (Mann-Whitney, two-tailed). Schematic representations of the assay conditions are shown at the top. **i** SIRF experiments showing that re-expression of wildtype PARP10 promotes the binding of RAD18 to ssDNA gaps induced by treatment with 0.4 mM HU in HeLa PARP10-knockout cells, but re-expression of the G888W catalytic site mutant or of the ΔPIP mutant does not. At least 72 cells were quantified for each condition. Bars indicate the mean values, error bars represent standard errors of the mean, and asterisks indicate statistical significance (*t*-test, two-tailed, unpaired). Schematic representations of the assay conditions are shown at the top. **j** Co-immunoprecipitation experiment in HeLa cells showing the interaction of PARP10 variants with RAD18. **k** PLA assay showing the co-localization between RAD18 and PARP10 variants upon treatment with 0.4 mM HU for 3hrs in HeLa cells. The catalytic site mutant (G888W) shows similar RAD18 co-localization as the wildtype form, while the PCNA interaction-deficient (ΔPIP) mutant shows reduced RAD18 co-localization. At least 75 cells were quantified for each condition. Bars indicate the mean values, error bars represent standard errors of the mean, and asterisks indicate statistical significance (*t*-test, two-tailed, unpaired). **l** SIRF experiments showing that re-expression of wildtype PARP10 promotes the binding of REV1 to ssDNA gaps induced by treatment with 0.4 mM HU in HeLa PARP10-knockout cells, but re-expression of the G888W catalytic site mutant or of the ΔPIP mutant does not. At least 72 cells were quantified for each condition. Bars indicate the mean values, error bars represent standard errors of the mean, and asterisks indicate statistical significance (*t*-test, two-tailed, unpaired). Schematic representations of the assay conditions are shown at the top. Source data are provided as a Source Data file.

(Fig. 6k). Importantly, BRCA2-depleted HeLa cells showed reduced proliferation upon treatment with OUL35, even in the absence of exposure to exogenous DNA damaging agents (Fig. 6l). This was accompanied by increased DSB formation as measured using the neutral comet assay (Fig. 6m; Supplementary Fig. 10a), in line with previous literature showing that inhibition of gap filling results in DSBs[7,8,60,62]. We next expanded the sensitivity studies to additional cell lines. BRCA2 depletion in 8988 T and RPE1 cells also resulted in reduced proliferation upon treatment with OUL35 in the absence of exposure to exogenous DNA damaging agents (Supplementary Fig. 10b, c). Moreover, BRCA2-depleted U2OS and DLD1 cells showed increased sensitivity to co-treatment with cisplatin and OUL35 compared to control cells (Supplementary Fig. 11a, b). Overall, these findings suggest that PARP10 suppresses ssDNA gap accumulation in BRCA-deficient cells and promotes their survival.

## Discussion

Our work identifies PARP10 as regulator of ssDNA gap filling. We show that PARP10 depletion causes accumulation of replication-dependent ssDNA gaps upon exposure to replication stress. PARP10 itself localizes to nascent DNA under gap-inducing conditions and interacts with RAD18, the main ubiquitin ligase responsible for PCNA ubiquitination. PARP10 depletion reduces RAD18 recruitment to nascent DNA under gap-inducing conditions, resulting in lower PCNA ubiquitination at these sites, and defects in recruitment of the TLS polymerase REV1 to nascent DNA under these conditions. Based on these findings, we speculate that PARP10 suppresses ssDNA gap accumulation through TLS polymerase-mediated gap filling. Indeed, we found that PARP10 is epistatic with RAD18 for gap suppression.

While HU and cisplatin cause replication stress through different mechanisms, previous research has shown that they both cause ssDNA gaps[8,18,23,25,56,57]. Loss of PARP10 caused ssDNA gap accumulation under both conditions, as shown using two orthogonal approaches to measure gap accumulation, namely the BrdU alkaline comet assay and the S1 nuclease DNA fiber combing assay, thus arguing for a general role for PARP10 in gap suppression. While statistically significant, the impact of PARP10 depletion is at times subtle, though it is comparable to that observed by depleting BRCA2 (Fig. 1). We also observed small differences in

the impact of PARP10 depletion on RAD18 recruitment and gap suppression between HU and cisplatin, likely reflecting the different ways in which they induce replication stress.

We confirmed the interaction between PARP10 and RAD18 using two orthogonal approaches, namely co-immunoprecipitation and PLA. The PLA assay, as well as the PLA-based SIRF assay are robust, quantitative assays that allow the quantification of changes in co-localization. However, a downside of these assays is that they may have a higher background signal. While we always confirmed the specificity of the signal by knocking down the proteins investigated, the reduction in the SIRF signal in these control samples is not dramatic, and in some cases only a 2-fold reduction was observed, pointing out that the signal to noise ratio may be relatively low.

Our findings indicate that PARP10's mono-ADP-ribosyltransferase enzymatic activity, as well as its PCNA interaction domain are required for this. Since the PIP-box motif is located at the beginning of the catalytic domain, we cannot rule out the possibility that the PIP-box mutation may possibly affect PARP10's catalytic activity. However, the findings that the PIP-box mutant shows reduced interaction with RAD18 while the catalytic site mutant shows normal RAD18 interaction (Fig. 4k) indicate that the two mutations are not equivalent and argues against the possibility that the observed phenotypes of the PIP-box mutant are caused by deficient catalytic activity. Instead, we speculate that PARP10 may be recruited to PCNA molecules at arrested forks, through its PIP-box motif. Our PLA and co-immunoprecipitation experiments suggest that PARP10 interacts with RAD18 to recruit it to its target PCNA at those locations. The catalytic activity of PARP10 may be involved in stabilizing these interactions. Indeed, we show that RAD18 can be mono-ADP-ribosylated by PARP10 in vitro. Moreover, we observed a specific PLA signal between RAD18 and MAR antibodies, which was dependent on PARP10. While we cannot rule out that the mono-ADP-ribosylation signal detected in these PLA experiments represents MARylation of RAD18-interacting proteins rather than of RAD18 itself, these results, together findings that RAD18 recruitment to ssDNA gaps is defective in the PARP10 catalytic site mutant, suggest that PARP10 MARylates RAD18 to promote its recruitment to gaps to initiate PCNA ubiquitination-dependent gap suppression.

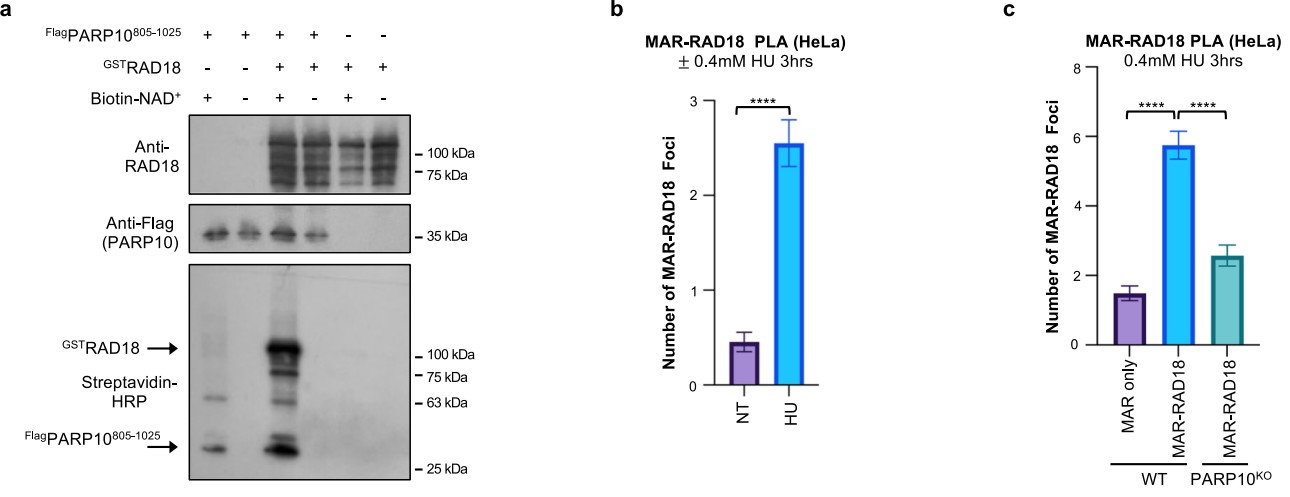

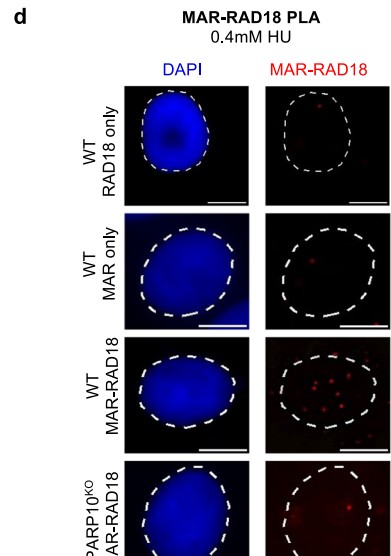

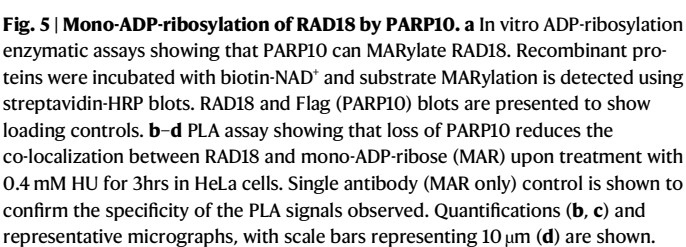

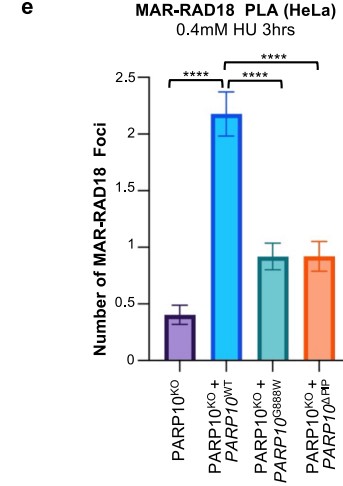

**Fig. 5 | Mono-ADP-ribosylation of RAD18 by PARP10. a** In vitro ADP-ribosylation enzymatic assays showing that PARP10 can MARylate RAD18. Recombinant proteins were incubated with biotin-NAD$^+$ and substrate MARylation is detected using streptavidin-HRP blots. RAD18 and Flag (PARP10) blots are presented to show loading controls. **b–d** PLA assay showing that loss of PARP10 reduces the co-localization between RAD18 and mono-ADP-ribose (MAR) upon treatment with 0.4 mM HU for 3hrs in HeLa cells. Single antibody (MAR only) control is shown to confirm the specificity of the PLA signals observed. Quantifications (**b, c**) and representative micrographs, with scale bars representing 10 μm (**d**) are shown.

At least 49 cells were quantified for each condition. Bars indicate the mean values, error bars represent standard errors of the mean, and asterisks indicate statistical significance (*t*-test, two-tailed, unpaired). **e** PLA assay showing the impact of PARP10 mutations on the co-localization between RAD18 and mono-ADP-ribose. Expression of wildtype PARP10, but not of the G888W catalytic site mutant or the PIP-box mutant restored the MAR-RAD18 PLA signal in PARP10$^{KO}$ cells. At least 60 cells were quantified for each condition. Bars indicate the mean values, error bars represent standard errors of the mean, and asterisks indicate statistical significance (*t*-test, two-tailed, unpaired). Source data are provided as a Source Data file.

RAD18 was previously shown to be recruited to stalled replication forks by interacting with RPA[63]. We speculate that this mode of recruitment is not affected by loss of PARP10. Indeed, in our previous studies, using western blot-based detection of ubiquitinated PCNA, we found that the impact of PARP10 loss on overall PCNA ubiquitination levels is less severe than that of RAD18 depletion[40,41]. While the difference is less evident in the $^{Ub}$PCNA SIRF experiments presented in this manuscript, this trend is still observable. We propose that PARP10 represents an additional mode of RAD18 recruitment, which may become more important under certain conditions such as gap filling (Fig. 7).

Studies over the past years have shown that BRCA-deficient cells accumulate ssDNA gaps, and gap accumulation correlates with their sensitivity to genotoxic agents[5,8,16–28,64]. Recent studies using separation of function BRCA2 mutants had however suggested that BRCA2 promotes chemotherapy resistance primarily through HR, rather than gap suppression[65,66]. PARP10-knockout cells are not hypersensitive to cisplatin, even though they accumulate gaps under those conditions. This suggests that, as long as cells are able to repair the gaps through BRCA-mediated recombination, their accumulation is not cytotoxic.

We propose that PARP10 depletion impairs the ability of cell to fill gaps using TLS polymerases. In BRCA-proficient cells, the gaps are

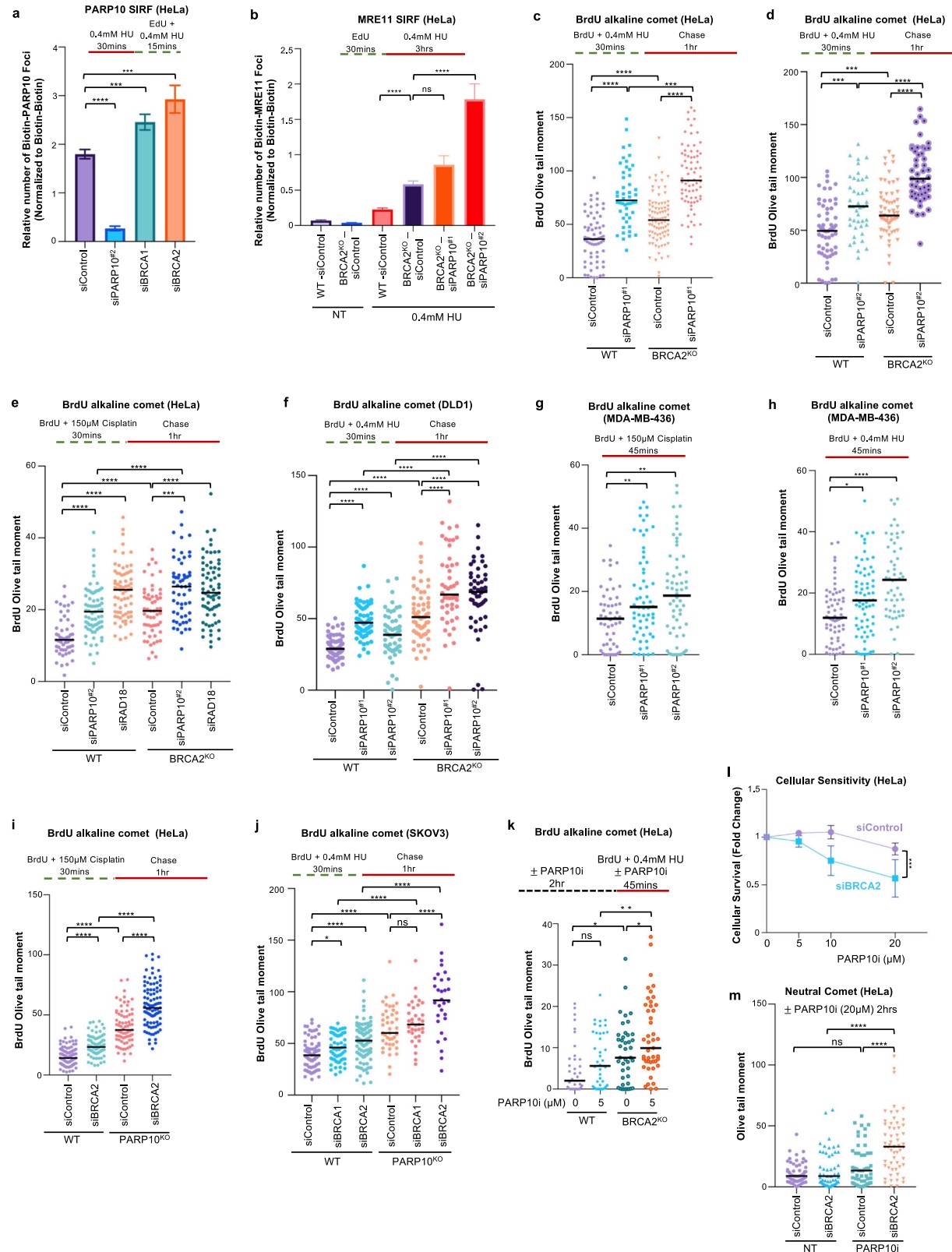

eventually fixed using homology-based repair. In BRCA-deficient cells, however, this repair is compromised, and further loss of PARP10 also reduces TLS-mediated gap filling. This renders cells with concomitant inactivation of BRCA and PARP10 unable to repair gaps, which are then exonucleolytic expanded by nuclease such as MRE11 eventually generating unrepairable DNA damage structures (Fig. 7). Interestingly,

even in the absence of DNA damaging treatment, BRCA2-depleted cells were sensitive to PARP10 inhibition. This is in line with previous studies showing that REV1 depletion reduced the viability of BRCA-deficient cells[7]. However, it is also possible that other functions of PARP10 may contribute to this increased sensitivity. Overall, our studies uncovered PARP10 as a potential target for treatment of BRCA-mutant tumors.

**Fig. 6 | Inactivation of PARP10 enhances gap accumulation and causes cytotoxicity in BRCA-deficient cells. a** SIRF assay showing increased recruitment of PARP10 to nascent DNA upon induction of ssDNA gaps by treatment with 0.4 mM HU in HeLa cells depleted of BRCA1 or BRCA2. At least 80 cells were quantified for each condition. Bars indicate the mean values, error bars represent standard errors of the mean, and asterisks indicate statistical significance (*t*-test, two-tailed, unpaired). A schematic representation of the assay conditions is shown at the top. Western blots confirming knockdowns of BRCA1 and BRCA2 are shown in Supplementary Fig. 1f, g. **b** SIRF assay showing increased recruitment of MRE11 to nascent DNA upon induction of ssDNA gaps by treatment with 0.4 mM HU in HeLa-BRCA2$^{KO}$ cells upon depletion of PARP10 and BRCA2. At least 39 cells were quantified for each condition. Bars indicate the mean values, error bars represent standard errors of the mean, and asterisks indicate statistical significance (*t*-test, two-tailed, unpaired). A schematic representation of the assay conditions is shown at the top. **c-f.** BrdU alkaline comet assays showing that depletion of PARP10 enhances the accumulation of ssDNA gaps in BRCA2-knockout HeLa (**c–e**) and DLD1 (**f**) cells upon treatment with 0.4 mM HU (**c, d, f**) or 150 μM cisplatin (**e**). At least 44 nuclei were quantified for each condition. The median values are marked on the graph and listed at the top. Asterisks indicate statistical significance (Mann-Whitney, two-tailed). Schematic representations of the assay conditions are shown at the top. **g,h.** BrdU alkaline comet assays showing that depletion of PARP10 enhances the accumulation of ssDNA gaps in BRCA1-mutant MDA-MB-436 cells upon treatment with 150 μM cisplatin (**g**) or 0.4 mM HU (**h**). At least 52 nuclei were quantified for each condition. The median values are marked on the graph and

listed at the top. Asterisks indicate statistical significance (Mann-Whitney, two-tailed). Schematic representations of the assay conditions are shown at the top. **l, j** BrdU alkaline comet assays showing that depletion of BRCA1 or BRCA2 enhances the accumulation of ssDNA gaps in PARP10-knockout HeLa (**i**) and SKOV3 (**j**) cells upon treatment with 150 μM cisplatin (**i**) or 0.4 mM HU (**j**). At least 30 nuclei were quantified for each condition. The median values are marked on the graph and listed at the top. Asterisks indicate statistical significance (Mann-Whitney, two-tailed). Schematic representations of the assay conditions are shown at the top. Western blots confirming PARP10 knockout in SKOV3 cells are shown in Supplementary Fig. 1h. **k** BrdU alkaline comet assay showing that PARP10 inhibition using the specific inhibitor OUL35 increases the accumulation of replication-associated ssDNA gaps upon treatment with 0.4 mM HU in HeLa-BRCA2$^{KO}$ cells. At least 38 nuclei were quantified for each condition. The median values are marked on the graph and listed at the top. Asterisks indicate statistical significance (Mann-Whitney, two-tailed). A schematic representation of the assay conditions is shown at the top. **l** Cellular viability assays showing that PARP10 inhibition using the specific inhibitor OUL35 causes cytotoxicity in BRCA2-depleted HeLa cells. The average of three independent experiments, with standard deviations indicated as error bars, is shown. Asterisks indicate statistical significance (two-way ANOVA). **m** Neutral comet assay showing that treatment of BRCA2-knockdown HeLa cells with OUL35 results in increased DSB formation. At least 50 comets were quantified for each sample. The median values are marked on the graph, and asterisks indicate statistical significance (Mann-Whitney, two-tailed). Source data are provided as a Source Data file.

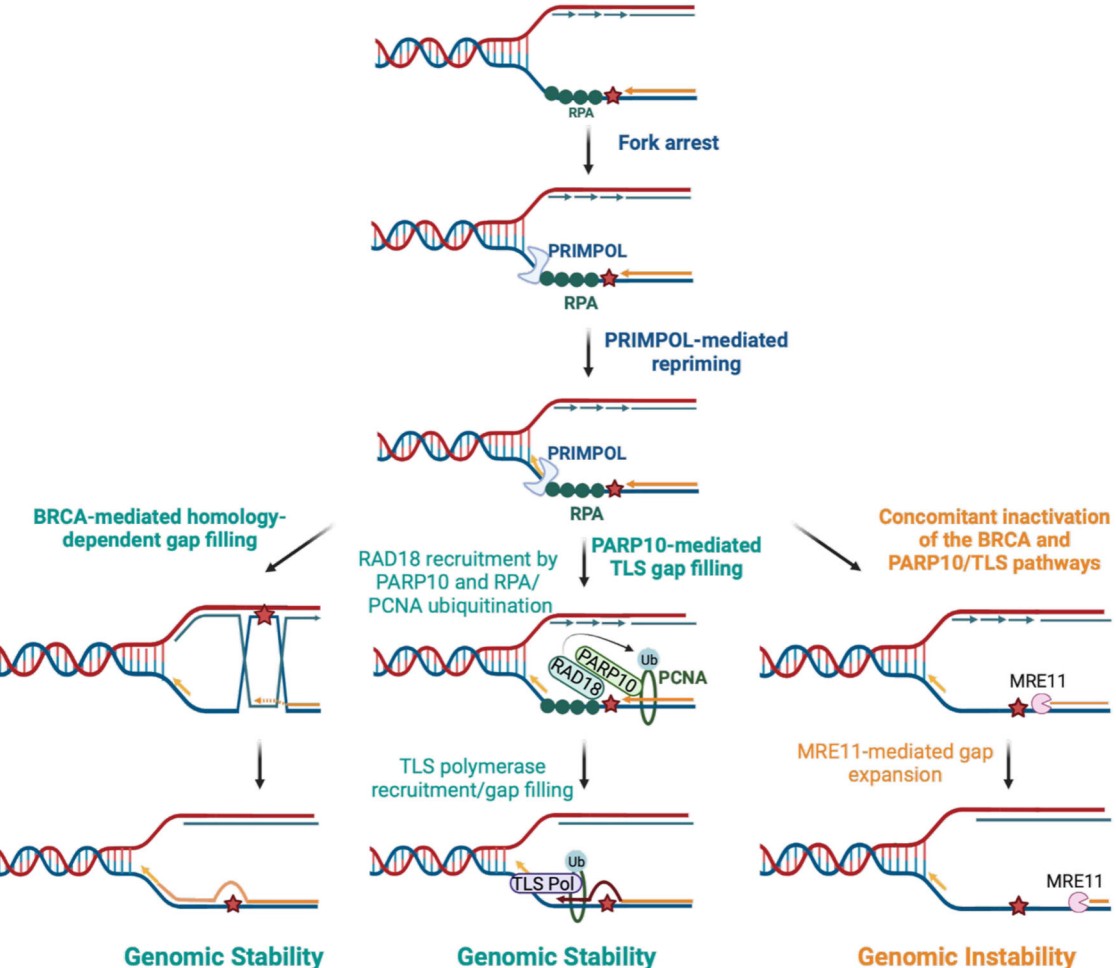

**Fig. 7 | Schematic representation of the proposed models.** PRIMPOL-derived ssDNA gaps can be filled through BRCA-mediated homology-dependent repair, or through PCNA ubiquitination-mediated TLS. Our work shows that PARP10 promotes the recruitment of RAD18, the main ubiquitin ligase for PCNA. Upon PCNA ubiquitination, TLS polymerases such as REV1 are recruited for gap filling. In cells with concomitant inactivation of BRCA and PARP10, ssDNA gaps are expanded by the MRE11 exonuclease, resulting in cytotoxicity. Created with BioRender.com released under a Creative Commons Attribution-NonCommercial-NoDerivs 4.0 International license.

## Methods

### Cell culture and protein techniques

HeLa (ATCC CCL-2), SKOV3 (ATCC HTB-77), MDA-MB-436 (ATCC HTB-130) and U2OS (ATCC HTB-96) cells (obtained from ATCC), as well as RPE1 and 8988 T cells (obtained from Dr. Alan D'Andrea, Dana-Farber Cancer Institute, Boston, MA) were grown in Dulbecco's modified Eagle's media (DMEM). DLD-1 and DLD1-BRCA2^KO cells (Horizon HD105-007) were obtained from Dr. Robert Brosh (National Institute on Aging, Baltimore, MD) and were grown in Roswell Park memorial Institute (RPMI) 1640 media. Media was supplemented with 15% FBS and penicillin/streptomycin. HeLa-BRCA2^KO cells were generated in our laboratory and previously described[67]. To knock-out PARP10 and RAD18, commercially available CRISPR/Cas9 KO plasmids (Santa Cruz Biotechnology sc-406703 and sc-406099 respectively) were used. Transfected cells were FACS-sorted into 96-well plates using a BD FACSAria II instrument. Resulting colonies were screened by Western blot. For re-expression of PARP10 wildtype and mutant variants, the pLV[Exp]-Puro-SV40 > hPARP10 lentiviral constructs (Cyagen) were used.

**Gene knockdown was performed using Lipofectamine RNAiMAX.** AllStars Negative Control siRNA (Qiagen 1027281) was used as control. The following oligonucleotide sequences (Stealth or SilencerSelect siRNA, ThermoFisher) were used: PARP10#1: GCCUGGUGGAGAUGGUG CUAUUGAU; PARP10#2: UGAAGGACCGGAUAUGACUGGCUUU; BRCA1: AAUGAGUCCAGUUUCGUUGCCUCUG; BRCA2: AUUAGGAGAAGACAUCAGAAGCUUG; RAD18: Assay ID s32295; USP1: Assay ID s14724; REV1: GAAAUCCUUGCAGAGACCAAACUUA; FEN1: Assay ID s5104; SMARCAL1: CACCCUUUGCUAACCCAACUCAUAA; ZRANB3: UGGCAAUGUAGUCUCUGCACCUAUA; RAD51: CCAUACUGUGGAGGCUGUUGCCUAU.

Denatured whole cell extracts were prepared by boiling cells in 100 mM Tris, 4% SDS, 0.5 M β-mercaptoethanol. Antibodies used for Western blot, at 1:500 dilution, were: PARP10: Abcam ab70800; RAD18: Cell Signaling Technology 9040;USP1: Abcam ab264221; BRCA1 Santa Cruz Biotechnology sc-6954; BRCA2 Bethyl A303-434A; FEN1: Santa Cruz Biotechnology sc-28355; SMARCAL1: Invitrogen PA5-54181; ZRANB3: Invitrogen PA5-6514; RAD51: Santa Cruz Biotechnology sc-8349; Streptavidin-HRP: ThermoFisher 21130; FLAG: Cell Signaling Technology 14793; Myc: Santa Cruz Biotechnology sc-40; Vinculin: Santa Cruz Biotechnology sc-73614; GAPDH: Santa Cruz Biotechnology sc-47724.

Chemical compounds used were: mirin (Selleck Chemicals S8096), OUL35 (Tocris Bioscience 6344).

### Functional assays

Neutral and BrdU alkaline comet assays were performed[28] using the Comet Assay Kit (Trevigen, 4250-050). For the BrdU alkaline comet assay, cells were incubated with 100 μM BrdU as indicated. Chemical compounds (HU, cisplatin, OUL35) were added according to the labeling schemes presented. Slides were stained with anti-BrdU (BD 347580) antibodies and secondary AF568-conjugated antibodies (Invitrogen A-11031). Slides were imaged on a Nikon microscope operating the NIS Elements V1.10.00 software. Olive tail moment was analyzed using CometScore 2.0.

### Drug sensitivity assays

To assess cellular viability upon drug treatment, a luminescent ATP-based assay was performed using the CellTiterGlo reagent (Promega G7572) according to the manufacturer's instructions. Following treatment with siRNA, 1500 cells were seeded per well in 96-well plates and incubated as indicated for 3 days. Luminescence was quantified using a Promega GloMax Navigator plate reader.

### DNA fiber combing assays

Cells were incubated with 100 μM IdU and 100 μM CldU as indicated. Chemical compounds (HU, cisplatin, mirin) were added according to the labeling schemes presented. Next, cells were collected and processed using the FiberPrep kit (Genomic Vision EXT-001) according to the manufacturer's instructions. Samples were added to combing reservoirs containing MES solution (2-(N-morpholino) ethanesulfonic acid) and DNA molecules were stretched onto coverslips (Genomic Vision COV-002-RUO) using the FiberComb Molecular Combing instrument (Genomic Vision MCS-001). For S1 nuclease assays, MES solution was supplemented with 1 mM zinc acetate and either 40 U/mL S1 nuclease (ThermoFisher 18001016) or S1 nuclease dilution buffer as control, and incubated for 30 min at room temperature. Slides were then stained with antibodies detecting CldU (Abcam 6236) and IdU (BD 347580), and incubated with secondary Cy3 (Abcam 6946) or Cy5 (Abcam 6565) conjugated antibodies. Finally, the cells were mounted onto coverslips and imaged using a confocal microscope (Leica SP5) and analyzed using LASX 3.5.7.23225 software. The scale bars for the DNA combing micrographs shown represent 10 μm.

### Proximity ligation-based assays

For PLA assays, cells were seeded into 8-chamber slides and 24 h later, were treated with 0.4 mM HU for 3 hrs as indicated. Cells were then permeabilized with 0.5% Triton for 10 min at 4 °C, washed with PBS, fixed at room temperature with 3% paraformaldehyde in PBS for 10 min, washed again in PBS and then blocked in Duolink blocking solution (Millipore Sigma DUO82007) for 1 hr at 37 °C, and incubated overnight at 4 °C with primary antibodies. Antibodies used were: PARP10 (Abcam ab70800); MAR AbD33204 (BioRad HCA354) and RAD18 (Cell Signaling Technology 9040). Samples were then subjected to a proximity ligation reaction using the Duolink kit (Millipore Sigma DUO92008) according to the manufacturer's instructions. Slides were imaged using a Deltavision microscope with SoftWorx 6.5.2 software, and images were analyzed using ImageJ 1.53a software.

For SIRF assays, cells were seeded into 8-chamber slides and 24 h later they were pulse-labeled with 50 μM EdU and treated with chemical compounds (HU, cisplatin, OUL35) according to the labeling schemes presented. Cells were permeabilized with 0.5% Triton for 10 min at 4 °C, washed with PBS, fixed at room temperature with 3% paraformaldehyde in PBS for 10 min, washed again in PBS, and then blocked in 3% BSA in PBS for 30 min. Cells were then subjected to Click-iT reaction with biotin-azide using the Click-iT Cell Reaction Buffer Kit (Thermo-Fisher C10269) for 30 min and incubated overnight at 4 °C with primary antibodies diluted in PBS with 1% BSA. The primary antibodies used were: Biotin (mouse: Jackson ImmunoResearch 200-002-211; rabbit: Bethyl Laboratories A150-109A); MRE11 (GeneTex GTX70212); PARP10 (Abcam ab70800); RAD18 (Cell Signaling Technology 9040); Ubiquityl-PCNA Lys164 (Cell Signaling Technology 13439); REV1 (Santa Cruz Biotechnology sc-393022); Myc (Santa Cruz Biotechnology sc-40). Next, samples were subjected to a proximity ligation reaction using the Duolink kit (MilliporeSigma DUO92008) according to the manufacturer's instructions. Slides were imaged using a Deltavision microscope with SoftWorx 6.5.2 software, and images were analyzed using ImageJ 1.53a software. To account for variation in EdU uptake between samples, for each sample, the number of protein-biotin foci were normalized to the average number of biotin-biotin foci for that respective sample. The scale bars for the SIRF and PLA micrographs shown represent 10 μm.

### Co-immunoprecipitation

Cells were lysed in HEPES lysis buffer (50 mM HEPES, 150 mM NaCl, 1 mM EDTA, 1% TritonX-100, 10% glycerol, 10 μM MgCl$_2$) supplemented with cOmplete Protease Inhibitor Cocktail (Roche 11836170001) for 30 min at 4 °C. Extracts were cleared by centrifugation and incubated with 2 μg anti-RAD18 (Cell Signaling Technology 9040) or control rabbit IgG (GenScript A01008) antibodies overnight at 4 °C, followed by incubation with Protein A/G PLUS-agarose beads (Santa Cruz

Biotechnology sc-2003) for 2 h at 4 °C. Beads were washed 5 times with HEPES lysis buffer, and eluted by boiling in Laemmli buffer.

## PARP10 enzymatic assays

For in vitro ADP-ribosylation of RAD18 by PARP10, commercially-available recombinant proteins were used. 1 μg PARP10[805-1025] (BPS Bioscience 80522) and/or 1 μg RAD18 (Abcam ab112417) were incubated with 25 μM biotin-NAD+ (BPS Bioscience 80610) in PARP assay buffer (BPS Bioscience 80602) for 2 h at room temperature. The reaction was stopped by boiling in Laemmli buffer and analyzed by western blot with Streptavidin-HRP antibodies (ThermoFisher 21130).

## Statistics and reproducibility

For SIRF and PLA assays the $t$-test (two-tailed, unpaired) was used. For DNA fiber assays and comet assays the Mann-Whitney statistical test (two-tailed) was performed. For CellTiterGlo cellular viability assays the two-way ANOVA statistical test or the $t$-test (two-tailed, unpaired) were used as indicated. For DNA fiber combing, PLA, SIRF, and comet assays, results from one experiment are shown; the results were reproduced in at least one additional independent biological conceptual replicate. Western blot and co-immunoprecipitation experiments were reproduced at least two times. Western blot image quantifications were performed with ImageJ.JS v0.5.8 run in browser (https://ij.imjoy.io). Statistical analyses were performed using GraphPad Prism 10 and Microsoft Excel v2205 software. Statistical significance is indicated for each graph (ns = not significant, for $p > 0.05$; * for $p \leq 0.05$; ** for $p \leq 0.01$; *** for $p \leq 0.001$, **** for $p \leq 0.0001$).

## Reporting summary

Further information on research design is available in the Nature Portfolio Reporting Summary linked to this article.

## Data availability

All data supporting the findings in this study are provided within this paper and its Supplementary Information file. Source data are provided with this paper.

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

## Acknowledgements

We would like to thank Grace Huang for technical support, Dr. Robert Brosh and Dr. Alan D'Andrea for materials, and the Penn State College of Medicine Advanced Light Microscopy (RRID:SCR_022526) and Flow Cytometry (RRID:SCR-021134) core facilities. Schematic figures were created with Biorender.com. This work was supported by: NIH R01CA244417 (to CMN), NIH R01ES026184 (to GLM) and NIH R01GM134681 (to GLM), as well as the Four Diamonds Transformative Patient-Oriented Cancer Research Project 4D01_2024_1002 (to GLM and CMN). The content is solely the responsibility of the authors and does not necessarily represent the official views of Four Diamonds.

## Author contributions

J.B.K., A.D., G.L.M. and C.M.N. designed and conducted the experiments, and wrote the paper.

## Competing interests

The authors declare no competing interests.
