## [Peer Review File · Nature Communications]

PARP10 promotes the repair of nascent strand DNA gaps through RAD18 mediated Translesion SynthesisREVIEWER COMMENTS

Reviewer #1 (Remarks to the Author):

Summary:

In line with previous work, the Nicolae and Moldovan laboratories show the involvement of PARP10 in processes related to DNA replication, namely DNA replication stress response and PCNA-mediated translesion synthesis (TLS).

Previously, the authors published the biochemical interaction of PARP10 with PCNA via its PIP-box motif. In addition to the PIP-box, the two PARP10 ubiquitin-interacting motifs (UIMs) participate in the PARP10-PCNA interaction by binding ubiquitinated PCNA, thus strengthening the interaction.

In this manuscript, Nicolae and Moldovan laboratories further characterised the role of PARP10 in DNA replication stress response. In detail, the authors show that PARP10 is part of the ssDNA gap filling process and activates TLS through a still uncharacterised PARP10-dependent regulation of PCNA and RAD18 localisation and functions.

PARP10 localises to nascent DNA under gap-inducing conditions, for instance under exposure to genotoxic agents, such as hydroxyurea (HU) and cisplatin. The recruitment of PARP10 to DNA gaps appears to depend on its ADP-ribosyl-transferase activity. Indeed, the chemical inhibitor of PARP10 reduced the PARP10 SIRF and led to the accumulation of ssDNA gaps upon treatment with HU or cisplatin.

At DNA gaps, PARP10 co-localised with RAD18, the main PCNA ubiquitin ligase, whose depletion resulted in similar phenotypes. Similarly, down-regulation of PARP10 impaired RAD18 recruitment to DNA gaps, suggesting that PARP10 promotes it. In turn, PARP10 depletion affected both RAD18-mediated PCNA ubiquitination and engagement of the TLS polymerase REV1.

Such observations were confirmed by rescue experiments in PARP10 KO cells, which were complemented with wild-type PARP10. By contrast, catalytically inactive PARP10, as well as PIP-box deletion mutant of PARP10, could not complement the observed phenotypes. These data, together with those obtained by treating cells with the chemical inhibition of PARP10, suggested that the catalytic activity of PARP10 is required for DNA gap synthesis, PCNA ubiquitination, and TLS.

PARP10-mediated gap suppression appears to be alternative to the BRCA1/2 mechanism. For such reason, BRCA1/2-depleted cells may greatly rely on PARP10. Indeed, PARP10 inhibition further improved sensitive of BRCA1/2-downregulated cells to DNA damaging agents.

Critique:

Although this work is scientifically valid and technically sound in methodology and statistical analysis, I found the current version of the manuscript to be too descriptive, with little progress compared with previous works published by the same laboratories. Several claims in this paper were overstated or unclear without providing molecular mechanisms. Therefore, it is not suitable for publication in Nature Communication in its current form.

1. One major issue of this paper is that the mechanism by which PARP10 controls PCNA ubiquitination via recruitment of RAD18 has not been described at the best. The authors showed that the catalytic activity of PARP10 is required for that; however it is not clear how it accomplishes this duty. Thus, the authors should clarify the following aspects:

- How does PARP10-dependent ADP-ribosylation stimulate localisation and functions of RAD18 and PCNA?

- How does PARP10 ADP-ribosyl transferase activity stimulate ubiquitination of PCNA? Are PCNA and/or RAD18 ADP-ribosylated by PARP10?

To address these points and further characterise the mechanisms, the authors should take advantage of recently developed antibodies to visualise PARylation, perhaps on immunoprecipitated proteins.

2. Other major issue of this paper concerns the demonstration of the physical interaction of PARP10 with both PCNA and RAD18 in cells.

PLA assays, which are nicely performed in this work, should be accompanied by reciprocal co-

immunoprecipitation studies on both endogenous and exogenous proteins (PARP10 WT and mutants), each one made under both DNA damaged and undamaged conditions.

3. Based on the domain organisation of PARP10, the PIP-motif appears to be positioned close to the catalytic domain.

- Does the PIP box mutation affect the enzymatic activity of PARP10? The authors should perform in vitro ADP-ribosylation assays to evaluate this aspect.

4. Does the mutation/deletion of UIM domain affect PARP10 localisation at DNA gaps?

Minor concerns:

1. All the experiments in Figure 2 should be performed in PARP10 KO cells as well

2. The authors should indicate the molecular weights for western blot images (e.g. Figure 4B and Supplementary File)

Reviewer #2 (Remarks to the Author):

Khatib et al

The submission to Nature Communications by Khatib and colleagues describe a role of the mono-ADP-ribosyltransferase PARP10 in ssDNA gap repair. They outline a series of events where PARP10 catalytic activity is required for RAD18 recruitment to ssDNA gaps, followed by RAD18-mediated PCNA ubiquitination, followed by REV1 gap filling. Furthermore, loss of PARP10 or inhibition exacerbates ssDNA gaps that arise in BRCA deficient cells and the latter are sensitive to an inhibitor of PARP10.

Taken together, the experiments are nicely performed and clearly presented. However, most of the data is descriptive, and limited mechanistic insights are provided that would warrant publication in Nature Communications.

Some suggestions that could improve the manuscript are as follows.

1. The authors show that PIP box and catalytic activity of PARP10 is required for RAD18 recruitment to ssDNA. However, they stop there. What is the mechanism of PARP10-mediated RAD18 recruitment?
2. The differences between comet and S1 assays are substantial. For example, comet gives almost 2-fold difference but S1 only 1.2-fold between PARP10 depletion and control. Does this suggest that PARP10 is doing more than inducing gaps on nascent strand?
3. 150uM cisplatin is significantly higher than concentration required to kill the majority of cell lines. Why is such a high concentration required? Is this physiologically relevant?
4. The mechanism by which PARP10 increases the gaps in BRCA2 cells is unclear. What is going on in this setting and the interplay with other factors described to suppress gaps in BRCA2 cells.

Reviewer #3 (Remarks to the Author):

In the present manuscript, Khatib and colleagues propose that PARP10 mono-ADP-ribosyltransferase is required to repair replication-associated ssDNA gaps via Translesion Synthesis. They showed that PARP10 suppresses ssDNA accumulation in Hydroxyurea and Cisplatin-treated cancer cells, requiring both its activity and interaction domain with PCNA. Moreover, they also showed that PARP10 mediates the recruitment of RAD18, impacting PCNA ubiquitination status, and REV1 polymerase to stalled replication forks. Finally, the authors gave evidence that BRCA-deficient cells utilize this mechanism for ssDNA gap repair as a salvage pathway when homology-based repair is compromised.

The manuscript is a direct continuation of previous works of the lab and the topic is highly interesting as it proposes a new regulator for ssDNA gap repair upon stressed forks. The finding that PARP10 is highly utilized in BRCA-deficient cancer cells is of particular interest as it may promote new therapeutic avenues for BRCA-deficient cancers. Generally, the experimental data is of high quality and supports most of the claims of the authors. However, several modifications and

important points need to be addressed in the manuscript prior publication.

Major points:

#1 – The manuscript is properly structured with a clear logic. However, some of the figures are unnecessarily dense and difficult to follow and I believe they should be compacted or simplified. Moreover, as all experiments are microscopy-based approaches, I believe it will be helpful to include example images to visualize what has been quantified in every experiment. This may be of particular importance for readers outside of the field.

#2 – As stated in Methods:

“For DNA fiber combing, PLA, SIF, and comet assays, results from one experiment are shown; the results were reproduced in at least one additional independent biological conceptual replicate. Western blot experiments were reproduced at least two times.”

If there is more biological data, it should be added to the figures. Importantly, it seems that the results are not as reproducible as expected (i.e., Figure 1H and Figure 5K show inconsistent results comparing BrdU Olive Tail Movement between WT and 5 μ M PARP10i). Another concern about the methods is the low amount of DNA combing tracks quantified (between 33 to 50 tracts). As this concern has been already brought in previous studies of the group (Hale et al., 2023), I believe it is necessary to increase the quantification of the combing data, at least to match the minimum of 50-90 tracts in previous studies, and add the biological replicates to strengthen the statistics.

#3 – The authors did not mention why experimental conditions for the same experiments change between figures, i.e.; Figure 1A-B and 4F (no thymidine chase), Figure 4D and 5A (incubation times change) or Figure 1J and 5B (different order of HU and length of pulse). Could the authors explain these changes in seemingly identical experiments?

#4 – Overall, the data is convincing about the role of PARP10 suppressing ssDNA accumulation at stalled replication forks. However, the model of PARP10 recruitment to stressed forks and subsequent interaction with RAD18 are only supported by SIF experiments. As this is a critical point of the manuscript, I believe an orthogonal approach is necessary to confirm this role of PARP10. Moreover, the authors observe in several cases that even an siRNA knock-down of their protein of interest leads to only a 2fold reduction of the SIF signal. A critical discussion of SIF and potential signal-to-noise issues would be appreciated.

#5 – The authors assume that HU and cisplatin have similar impact to replication forks, when they are two dissimilar sources of replication stress. A clear example is the S1 combing experiment (Figure 1E and 1F), where Cisplatin treatment has a CldU/IdU ratio of 2 whereas HU has a ratio of 1. Another example, SIF experiments showed a different fold decrease in signal of RAD18 in siPARP10 #2 conditions (Figures 2G and 2H). I believe this is worth mentioning, at least, in discussion of the data.

#6 – In my opinion, the sensitivity to PARP10i in BRCA deficient conditions is one of the most important results of the paper. As the authors generally use several cell lines to confirm the generality of their finding, I believe this should also be done for the cellular sensitivity assays.

#7 – The authors did not take into consideration that ssDNA signal from both S1 DNA combing and BrdU comet assays could be due to fork reversal and ssDNA accumulation in the reverse strand. Indeed, low doses of HU (0,5 mM) have been shown to promote fork reversal in U2OS and RPE-1 cells (>20% forks, Zellweger et al., 2015). This fact is critical as many of the differences are quite subtle, even though statistically relevant, and should be indicated in the manuscript as authors did in Hale et al., 2023.

Minor points:

- Line 111 and 112: Can the authors deduce physical interaction between PARP10 and RAD18 from PLA as stated here? See their much more nuanced discussion of their data (Line 344 - 346).

- Line 121: “PARP10 is required for suppressing the accumulation of ssDNA gaps in wildtype cells” Both HeLa and DLD1 cells are cancer-derived cell lines, thus no WT cells. Either change the title or perform experiments in non-cancerous cell lines.

- Line 201 and 202: PLA demonstrates proximity, not interaction. Rephrase as in discussion.

- Figure 2D: PARP10KO condition was treated with 4mM HU instead of 0.4 mM HU. Is it a typo?

- Line 207 – 209: it is unclear how this data suggests a feedback loop. I would like a further explanation.

- Figure 3A: ubPCNA SIF experiments are missing NT conditions.

- Line 234 – 235 and Figure 3H: missing stats comparing WT and RAD18 KO, as presented in Figure 3G.

- Line 235: HeLa cells are not WT cells, correct accordingly.

- Line 352 – 354: “Indeed, in our previous studies we found that the impact of PARP10 loss on overall PCNA ubiquitination levels is less severe than that of RAD18 depletion” In figure 3A it is not evident. Please provide stats to support that claim in this figure.

Response to referees

We would like to thank the reviewers for their helpful and constructive comments, which led to a significantly improved manuscript. To address the reviewers' concern, we are submitting a substantially revised manuscript with 32 new figure panels, as well as 6 revised figure panels.

Reviewer #1 (Remarks to the Author):

Summary:

In line with previous work, the Nicolae and Moldovan laboratories show the involvement of PARP10 in processes related to DNA replication, namely DNA replication stress response and PCNA-mediated translesion synthesis (TLS).

Previously, the authors published the biochemical interaction of PARP10 with PCNA via its PIP-box motif. In addition to the PIP-box, the two PARP10 ubiquitin-interacting motifs (UIMs) participate in the PARP10-PCNA interaction by binding ubiquitinated PCNA, thus strengthening the interaction.

In this manuscript, Nicolae and Moldovan laboratories further characterised the role of PARP10 in DNA replication stress response. In detail, the authors show that PARP10 is part of the ssDNA gap filling process and activates TLS through a still uncharacterised PARP10-dependent regulation of PCNA and RAD18 localisation and functions.

PARP10 localises to nascent DNA under gap-inducing conditions, for instance under exposure to genotoxic agents, such as hydroxyurea (HU) and cisplatin. The recruitment of PARP10 to DNA gaps appears to depend on its ADP-ribosyl-transferase activity. Indeed, the chemical inhibitor of PARP10 reduced the PARP10 SIF and led to the accumulation of ssDNA gaps upon treatment with HU or cisplatin.

At DNA gaps, PARP10 co-localised with RAD18, the main PCNA ubiquitin ligase, whose depletion resulted in similar phenotypes. Similarly, down-regulation of PARP10 impaired RAD18 recruitment to DNA gaps, suggesting that PARP10 promotes it. In turn, PARP10 depletion affected both RAD18-mediated PCNA ubiquitination and engagement of the TLS polymerase REV1.

Such observations were confirmed by rescue experiments in PARP10 KO cells, which were complemented with wild-type PARP10. By contrast, catalytically inactive PARP10, as well as PIP-box deletion mutant of PARP10, could not complement the observed phenotypes. These data, together with those obtained by treating cells with the chemical inhibition of PARP10, suggested that the catalytic activity of PARP10 is required for DNA gap synthesis, PCNA ubiquitination, and TLS.

PARP10-mediated gap suppression appears to be alternative to the BRCA1/2 mechanism. For such reason, BRCA1/2-depleted cells may greatly rely on PARP10. Indeed, PARP10 inhibition further improved sensitive of BRCA1/2-downregulated cells to DNA damaging agents.

Critique:

Although this work is scientifically valid and technically sound in methodology and statistical analysis, I found the current version of the manuscript to be too descriptive, with little progress

compared with previous works published by the same laboratories. Several claims in this paper were overstated or unclear without providing molecular mechanisms. Therefore, it is not suitable for publication in NatureCommunication in its current form.

We thank the reviewer for their very useful comments. We were glad that the reviewer found our work to be "scientifically valid and technically sound". We would like to respectively point out that the role of PARP10 in gap suppression described here, the interaction between PARP10 and RAD18, and the reliance of BRCA2-deficient cells on PARP10, are entirely novel findings and do not represent a simple extension or obvious follow-up of our previous work. In the revised manuscript, we further explored the mechanistic aspects of PARP10-mediated gap suppression, focusing on its interaction with RAD18, as requested by the reviewer.

1. One major issue of this paper is that the mechanism by which PARP10 controls PCNA ubiquitination via recruitment of RAD18 has not been described at the best. The authors showed that the catalytic activity of PARP10 is required for that; however it is not clear how it accomplishes this duty. Thus, the authors should clarify the following aspects:

- How does PARP10-dependent ADP-ribosylation stimulate localisation and functions of RAD18 and PCNA?

- How does PARP10 ADP-ribosyl transferase activity stimulate ubiquitination of PCNA? Are PCNA and/or RAD18 ADP-ribosylated by PARP10?

To address these points and further characterise the mechanisms, the authors should take advantage of recently developed antibodies to visualise MARYlation, perhaps on immunoprecipitated proteins.

We thank the reviewer for these helpful suggestions and comments. In the revised manuscript, we show that PARP10 mono-ADP-ribosylates RAD18 *in vitro* (**new Fig. 5a**). In contrast, we did not consistently observe ADP-ribosylation of PCNA (not shown). We also attempted extensively to employ the MARYlation antibodies to detect RAD18 modification on immunoprecipitated samples. However, we were not able to detect any specific signal using these antibodies in western blot. While the reason for this is unclear, we were able to use these antibodies in proximity ligation (PLA) assays. We were able to detect a specific signal when performing PLA experiments with anti-RAD18 and anti-mono-ADP-ribosylation (Anti-MAR) antibodies, which was specifically increased by replication stress exposure. This signal was reduced in PARP10-knockout cells (**new Fig. 5b-d**). Expression of wildtype PARP10, but not of the G888W catalytic site mutant or the PIP-box mutant restored the RAD18-MAR signal in PARP10^{KO} cells (**new Fig. 5e**). These findings indicate a PARP10-dependent co-localization of RAD18 and mono-ADP-ribosylation, increased under gap-inducing conditions. While we cannot rule out that the mono-ADP-ribosylation signal detected in these PLA experiments represents MARYlation of RAD18-interacting proteins rather than of RAD18 itself, these results, together with the demonstration of *in vitro* MARYlation of RAD18 by PARP10, and the fact that RAD18 recruitment to ssDNA gaps is defective in the PARP10 catalytic mutant, suggest that PARP10 MARYlates RAD18 to promote its recruitment to gaps to initiate PCNA ubiquitination-dependent gap suppression.

2. Other major issue of this paper concerns the demonstration of the physical interaction of PARP10 with both PCNA and RAD18 in cells.

PLA assays, which are nicely performed in this work, should be accompanied by reciprocal co-immunoprecipitation studies on both endogenous and exogenous proteins (PARP10 WT and

mutants), each one made under both DNA damaged and undamaged conditions.

We thank the reviewer for this comment. We extensively performed co-immunoprecipitation experiments under various conditions. We were unable to observe co-immunoprecipitation of RAD18 when pulling down PARP10, likely because of technical (antibody-related) reasons. However, we could successfully co-immunoprecipitate PARP10 when pulling down RAD18. When performing these experiments under undamaged and DNA damaged conditions, as instructed by the reviewer, we observed only a very minor, if any, increase in the interaction in HU and cisplatin-treated samples compared to untreated samples (**new Fig. 2d**). We believe this reflects the fact that co-immunoprecipitation is not a quantitative assay. The quantitative PLA assays shown in our original manuscript (**Figure 2**) clearly indicated that the interaction is increased by replication stress exposure.

We also performed the co-immunoprecipitation experiment with the PARP10 mutants, as requested by the reviewer. Also in these experiments, while it appeared that the mutants may have reduced interaction with PARP10, it was difficult to unequivocally conclude this (**new Fig. 4j**). To clarify this issue, we employed the PLA assay to measure these interactions. We found that the catalytic mutant G888W retains normal interaction with RAD18, while the PIP-mutant shows reduced interaction (**new Fig. 4k, Supplementary Fig. S5**). These findings suggest that PARP10 recruitment to PCNA promotes or stabilizes its interaction with RAD18, potentially through formation of a heterotrimeric RAD18-PARP10-PCNA complex (see model in **Figure 7**).

3. Based on the domain organisation of PARP10, the PIP-motif appears to be positioned close to the catalytic domain.

- Does the PIP box mutation affect the enzymatic activity of PARP10? The authors should perform in vitro ADP-ribosylation assays to evaluate this aspect.

Indeed, the PIP box is at the beginning of the PARP domain. In order to address the reviewer's comment, during the course of this revision, we attempted to purify a fragment of PARP10 spanning the PIP box and catalytic domain (805-1025). We cloned the fragments in a pET bacterial expression vector, with a His-tag, and induced its expression in BL21(DE3) protein expression-competent bacteria. We were able to successfully induce the expression of wildtype, PIP-box mutant, and catalytic site mutant G888W variants. However, when we attempted to purify these variants, we were able to solubilize the wildtype and G888W mutant in bacterial native extracts, while the PIP-box mutant accumulated in inclusion bodies and could not be solubilized. Therefore, we were unable to purify it during the course of this revision.

Unfortunately, because of this technical issue, we cannot assess its enzymatic activity. We unsuccessfully tried several induction conditions and lysis methods. We are presenting below a figure with representative results of our purification efforts.

In the revised manuscript, we include a statement (on page 17) that a caveat of our studies with the PIP-box mutant is that this mutation may possibly affect PARP10's catalytic activity, since it is located at the beginning of the catalytic domain. However, the findings that the PIP-box mutant shows reduced interaction with RAD18 while the catalytic site mutant shows normal RAD18 interaction (**new Fig. 4k**) indicates that the two mutations are not equivalent and argues against the possibility that the observed phenotypes of the PIP-box mutant are caused by deficient catalytic activity.

Left: Denatured bacterial extracts showing IPTG-induced expression of PARP10⁸⁰⁵⁻¹⁰²⁵ His-tagged variants.
 Right: Native extracts and NiNTA beads after incubation with the extracts, showing that the PIP-box mutant is not soluble.

4. Does the mutation/deletion of UIM domain affect PARP10 localisation at DNA gaps?

In the revised manuscript, we now show that deletion of the UIM domains does not reduce the localization of PARP10 to ssDNA gaps (**new Supplementary Fig. S4a,b**). This is, in fact, in line with our model that PARP10 recruitment to ssDNA gaps occurs prior to PCNA ubiquitination, and is required for efficient RAD18-mediated ubiquitination.

Minor concerns:

1. All the experiments in Figure 2 should be performed in PARP10 KO cells as well

We now present the PARP10 SIF in PARP10KO cells in the **new Supplementary Fig. S3a**. Moreover, we present the RAD18-PARP10 PLA in PARP10^{KO} cells in the **new Supplementary Fig. S3b and S5**. The RAD18 SIF in PARP10^{KO} cells was already presented in our original manuscript (Fig. 4i).

2. The authors should indicate the molecular weights for western blot images (e.g. Figure 4B and Supplementary File)

We have now included molecular weights for the western blots shown in the main figures (**Fig. 4b, new Fig. 2d, 4j, 5a**). The western blots in the supplementary figures have the molecular weights shown in the Source Data file, where the whole gel is presented (since it is impractical to show them in the cropped images presented in the supplementary figures).

Reviewer #2 (Remarks to the Author):

Khatib et al

The submission to Nature Communications by Khatib and colleagues describe a role of the mono-ADP-ribosyltransferase PARP10 in ssDNA gap repair. They outline a series of events where PARP10 catalytic activity is required for RAD18 recruitment to ssDNA gaps, followed by RAD18-mediated PCNA ubiquitination, followed by REV1 gap filling. Furthermore, loss of PARP10 or inhibition exacerbates ssDNA gaps that arise in BRCA deficient cells and the latter are sensitive to an inhibitor of PARP10.

Taken together, the experiments are nicely performed and clearly presented. However, most of the data is descriptive, and limited mechanistic insights are provided that would warrant publication in Nature Communications.

We thank the reviewer for their comments. We were glad that the reviewer found our work to be “*nicely performed and clearly presented*”. In the revised manuscript, we further explored the mechanistic aspects of PARP10-mediated gap suppression, focusing on its interaction with and recruitment of RAD18, as requested by the reviewer.

Some suggestions that could improve the manuscript are as follows.

1. The authors show that PIP box and catalytic activity of PARP10 is required for RAD18 recruitment to ssDNA. However, they stop there. What is the mechanism of PARP10-mediated RAD18 recruitment?

In the revised manuscript, we show that PARP10 mono-ADP-ribosylates RAD18 *in vitro* (**new Fig. 5a**). Moreover, we employed antibodies detecting mono-ADP-ribosylation (MAR) to detect RAD18 MARYlation in cells. Using proximity ligation (PLA) assays, we were able to detect a specific signal when performing PLA experiments with Anti-RAD18 and Anti-MAR antibodies, which was specifically increased by replication stress exposure. This signal was reduced in PARP10-knockout cells (**new Fig. 5b-d**). Expression of wildtype PARP10, but not of the G888W catalytic site mutant or the PIP-box mutant restored the RAD18-MAR signal in PARP10^{KO} cells (**new Fig. 5e**). These findings indicate a PARP10-dependent co-localization of RAD18 and mono-ADP-ribosylation, increased under gap-inducing conditions. While we cannot rule out that the mono-ADP-ribosylation signal detected in these PLA experiments represents MARYlation of RAD18-interacting proteins rather than of RAD18 itself, these results, together with the demonstration of *in vitro* MARYlation of RAD18 by PARP10, and the fact that RAD18 recruitment to ssDNA gaps is defective in the PARP10 catalytic mutant, suggest that PARP10 MARYlates RAD18 to promote its recruitment to gaps to initiate PCNA ubiquitination-dependent gap suppression.

2. The differences between comet and S1 assays are substantial. For example, comet gives almost 2-fold difference but S1 only 1.2-fold between PARP10 depletion and control. Does this suggest that PARP10 is doing more than inducing gaps on nascent strand?

The BrdU alkaline comet assay and the S1 nuclease DNA fiber combing assay are very different methods for measuring ssDNA gaps. The comet readout is based on DNA electrophoresis, while the combing assay measures individual replication tracts from immunofluorescence micrographs. Because of these distinctions, we do not believe that the differences in the signal between the comet and the combing approaches are relevant. The S1 nuclease assay is considered the more specific approach to measure ssDNA gaps, with most papers in the field employing only this approach. However, we are using both methods to validate our findings in an orthogonal manner. We now discuss this in the revised manuscript (page 16). We have no evidence that PARP10 loss may induce other types of lesions on nascent DNA detectable by the BrdU alkaline comet assays.

3. 150uM cisplatin is significantly higher than concentration required to kill the majority of cell lines. Why is such a high concentration required? Is this physiologically relevant?

Indeed, this concentration of cisplatin is very high. The cells are treated with this cisplatin dose only for 2 hours, and no viability issues are observed under these conditions. The issue of physiological relevance is an ongoing one in the field. We have used this concentration because previous studies by other laboratories established it as a ssDNA gap-inducing conditions (eg. PMID: 31676232, PMID: 34624216). We have not verified ourselves if gaps are also induced at lower doses. As indicated in our manuscript, 150uM is a standard cisplatin concentration used in the literature for measuring nascent strand gaps. However, to rule out that the findings only apply to one specific treatment condition, we validated our cisplatin results using a different condition previously shown to induce nascent strand ssDNA gaps, namely treatment with 0.4mM HU. As pointed out by reviewer 3, this treatment causes replication stress through a different mechanism than cisplatin. Our studies show that PARP10 promotes gap suppression in both conditions, arguing for a general role of PARP10 on gap suppression.

4. The mechanism by which PARP10 increases the gaps in BRCA2 cells is unclear. What is going on in this setting and the interplay with other factors described to suppress gaps in BRCA2 cells.

We apologize for not presenting this in a more clear manner in our original manuscript. Our studies show that PARP10 promotes ssDNA gap suppression through activating REV1-mediated translesion synthesis (TLS). In contrast, BRCA2 is known to suppress gaps through a different mechanism, namely homology-based repair using the nascent strand of the sister chromatid as a template. Loss of PARP10 increases ssDNA gaps in BRCA-deficient cells through the same mechanism as in wildtype cells, namely by suppressing TLS. In cells with concomitant inactivation of both TLS and BRCA-mediated HR, ssDNA gaps have reduced avenues of repair.

In the revised manuscript, we addressed the interplay with other factors involved in gap suppression in BRCA2-deficient cells, as requested by the reviewer. Loss of FEN1 was previously shown to suppress HU-induced ssDNA accumulation in BRCA2 deficient cells (PMID: 33184108). We found that loss of FEN1 does not impact ssDNA accumulation in PARP10-deficient cells (**new Supplementary Fig. S7a,b**). These findings are in line with our model that PARP10 and BRCA2 promote gap suppression through different mechanisms. To further validate this, we also depleted RAD51, which is the main effector of BRCA2-mediated repair. Loss of RAD51 in PARP10-deficient cells resulted in a further increase in gap accumulation under both 0.4mM HU and 150uM cisplatin treatment conditions (**new Supplementary Figure**

6a-c), similar to the concomitant loss of PARP10 and BRCA2. These findings further indicate that PARP10 and the BRCA pathway suppress gaps through different mechanisms.

Reviewer #3 (Remarks to the Author):

In the present manuscript, Khatib and colleagues propose that PARP10 mono-ADP-ribosyltransferase is required to repair replication-associated ssDNA gaps via Translesion Synthesis. They showed that PARP10 suppresses ssDNA accumulation in Hydroxyurea and Cisplatin-treated cancer cells, requiring both its activity and interaction domain with PCNA. Moreover, they also showed that PARP10 mediates the recruitment of RAD18, impacting PCNA ubiquitination status, and REV1 polymerase to stalled replication forks. Finally, the authors gave evidence that BRCA-deficient cells utilize this mechanism for ssDNA gap repair as a salvage pathway when homology-based repair is compromised.

The manuscript is a direct continuation of previous works of the lab and the topic is highly interesting as it proposes a new regulator for ssDNA gap repair upon stressed forks. The finding that PARP10 is highly utilized in BRCA-deficient cancer cells is of particular interest as it may promote new therapeutic avenues for BRCA-deficient cancers. Generally, the experimental data is of high quality and supports most of the claims of the authors. However, several modifications and important points need to be addressed in the manuscript prior publication.

We thank the reviewer for their comments. We were glad that the reviewer found our work to be “highly interesting” and “of high quality”. In the revised manuscript, we addressed the reviewer’s comments as detailed below.

Major points:

#1 – The manuscript is properly structured with a clear logic. However, some of the figures are unnecessarily dense and difficult to follow and I believe they should be compacted or simplified. Moreover, as all experiments are microscopy-based approaches, I believe it will be helpful to include example images to visualize what has been quantified in every experiment. This may be of particular importance for readers outside of the field.

In the revised manuscript, we attempted to streamline the manuscript. We moved some of the material to the Supplementary Information section. We also added images of micrographs for DNA fiber combing experiments (new Fig. 1e,g, 4h), and added additional micrograph images for PLA assays (new Fig. 5c). In addition, we added experiments with western blot readouts to complement our microscopy-based assays (new Fig. 2d, 4j, 5a).

#2 – As stated in Methods:

“For DNA fiber combing, PLA, SIRF, and comet assays, results from one experiment are shown; the results were reproduced in at least one additional independent biological conceptual replicate. Western blot experiments were reproduced at least two times.”

If there is more biological data, it should be added to the figures. Importantly, it seems that the results are not as reproducible as expected (i.e., Figure 1H and Figure 5K show inconsistent results comparing BrdU Olive Tail Movement between WT and 5 μM PARP10i). Another concern about the methods is the low amount of DNA combing tracks quantified (between 33 to 50 tracts). As this concern has been already brought in previous studies of the group (Hale et

al., 2023), I believe is necessary to increase the quantification of the combing data, at least to match the minimum of 50-90 tracts in previous studies, and add the biological replicates to strengthen the statistics.

In the revised manuscript, we increased the quantification of the combing experiments to at least 60 tracts for each sample (**updated Fig. 1d,f,i,k,4g**). In conceptual replicates presented throughout the manuscript, we have shown that PARP10 is required for gap suppression using both knockdown and knockout approaches, including complementation with wildtype and mutants, in response to both HU and cisplatin. Regarding the BrdU alkaline comet indicated by the reviewer, while the starting tail moment value can vary from experiment to experiment, we consistently observe a 2-3 fold increase in the BrdU Olive Tail Movement between WT and 5 μ M PARP10i.

#3 – The authors did not mention why experimental conditions for the same experiments change between figures, i.e.; Figure 1A-B and 4F (no thymidine chase), Figure 4D and 5A (incubation times change) or Figure 1J and 5B (different order of HU and length of pulse). Could the authors explain these changes in seemingly identical experiments?

In general, while there may be slight changes in labeling times from experiment to experiment (resulting from tinkering with the experimental conditions to improve the signal, throughout the 2-3 years it took to obtain this data), the labeling schemes (orders of compounds added) were consistent for similar experiments. This being said, the reviewer is correct in pointing out the discrepancies listed in their comment. Some of these were errors in figures labeling, which we have fixed as listed below. We apologize for this, and we thank the reviewer for pointing them out. In a few other cases, we had a slightly different labeling scheme for some of the experiments. We have now repeated those experiments (see below), so that we now provide similar labeling schemes for similar experiments, throughout the manuscript.

Figure 1A-B and 4F (no thymidine chase): We now replaced the original Figure 4F with a new figure which uses the same labeling scheme as Figure 1A-B (**new Fig. 4f**).

Figure 4D and 5A (incubation times change): We apologize for this, but we had made an error in the labeling of Figure 5A (Fig. 6a in the revised manuscript): The correct labeling is: 0.4mM HU for 30mins / EdU+ 0.4mM HU for 15mins. This is essentially the same labeling scheme as for the other PARP10 SIF experiments (eg Figure 2). Regarding Figure 4D, we replaced the original figure with a new experiment performed using a similar labeling scheme as in the other PARP10 SIF experiments (as indicated above) (**new Fig. 4d**)

Figure 1J and 5B (different order of HU and length of pulse): We apologize for this, but we had made an error in the labeling of Figure 5B (Fig. 6b in the revised manuscript). The correct labeling is the same as in the original Figure 1J (Fig. 1I in the revised manuscript): EdU 30mins / 0.4mM HU 3hrs. This labeling scheme has been used before by us and others to measure MRE11 engagement on nascent DNA, considering its 3'-5' exonuclease activity.

#4 – Overall, the data is convincing about the role of PARP10 suppressing ssDNA accumulation at stalled replication forks. However, the model of PARP10 recruitment to stressed forks and subsequent interaction with RAD18 are only supported by SIF experiments. As this is a critical point of the manuscript, I believe an orthogonal approach is necessary to confirm this role of PARP10. Moreover, the authors observe in several cases that even an siRNA knock-down of

their protein of interest leads to only a 2fold reduction of the SIRF signal. A critical discussion of SIRF and potential signal-to-noise issues would be appreciated.

In the revised manuscript, we include co-immunoprecipitation experiments as an orthogonal approach to demonstrate the interaction between RAD18 and PARP10 (new Fig. 2d, 4j). We also show that PARP10 mono-ADP-ribosylates RAD18 (new Fig. 5a-e). In the revised manuscript, we include a discussion of the SIRF assay limitations as indicated by the reviewer (pages 17).

#5 – The authors assume that HU and cisplatin have similar impact to replication forks, when they are two dissimilar sources of replication stress. A clear example is the S1 combing experiment (Figure 1E and 1F), where Cisplatin treatment has a CldU/IdU ratio of 2 whereas HU has a ratio of 1. Another example, SIRF experiments showed a different fold decrease in signal of RAD18 in siPARP10 #2 conditions (Figures 2G and 2H). I believe this is worth mentioning, at least, in discussion of the data.

We agree with the reviewer that that HU and cisplatin impact fork progression through different mechanisms. In fact, this is the reason why we used both of them: to confirm that PARP10 promotes gap suppression under both conditions, thus arguing for a general role of PARP10 on gap suppression, rather than a role restricted to a unique source of damage/stress. While we agree with the reviewer that there are differences in the PARP10 response between HU and cisplatin, overall our results indicate that PARP10 promotes gap suppression under both conditions. We now address this issue in the Discussion section, as indicated by the reviewer (page 16).

#6 – In my opinion, the sensitivity to PARP10i in BRCA deficient conditions is one of the most important results of the paper. As the authors generally use several cell lines to confirm the generality of their finding, I believe this should also be done for the cellular sensitivity assays.

We thank the reviewer for this helpful comment. In the revised manuscript, we extended the analyses to 8988T and RPE1 cell lines, to show that BRCA2-knockdown cells have increased sensitivity to PARP10i (new Supplementary Fig. S8a,b). In addition, we employed U2OS and DLD1 cell lines. We show that, in these cell lines, BRCA2 depletion results in increased sensitization upon co-treatment with cisplatin and PARP10i (new Supplementary Fig. S9a,b). Finally, to provide a mechanism for the observed sensitivity, we now show that PARP10i-treated BRCA2 knockdown HeLa cells accumulate DSBs (new Fig. 6m), in line with previous literature indicating that inhibition of gap filling leads to their conversion into DSBs.

7 – The authors did not take into consideration that ssDNA signal from both S1 DNA combing and BrdU comet assays could be due to fork reversal and ssDNA accumulation in the reverse strand. Indeed, low doses of HU (0,5 mM) have been shown to promote fork reversal in U2OS and RPE-1 cells (>20% forks, Zellweger et al., 2015). This fact is critical as many of the differences are quite subtle, even though statistically relevant, and should be indicated in the manuscript as authors did in Hale et al., 2023.

We thank the reviewer for this comment. In the revised manuscript, we show that depletion of fork reversal translocases ZRANB3 and SMARCA1 does not affect HU-induced ssDNA gap accumulation in PARP10-knockout cells (new Supplementary Fig. S2a-c), arguing that the

gaps do not occur on reversed forks. We agree with the reviewer that some of the difference observed may be at times subtle, but we would like to point out that in general, the impact of PARP10 depletion on gap accumulation is comparable to that of BRCA2 depletion. Thus, we believe that the fact that the differences are subtle reflects the technical challenges of the assays, rather than a reduced physiological relevance of PARP10 in gap suppression. We addressed this issue in the Discussion section, as requested by the reviewer (pages 16-17)

Minor points:

- Line 111 and 112: *Can the authors deduce physical interaction between PARP10 and RAD18 from PLA as stated here? See their much more nuanced discussion of their data (Line 344 - 346).*

As mentioned above, in the revised manuscript, in addition to PLA assays, we present co-immunoprecipitation experiments to support the claim that PARP10 interacts with RAD18 (**new Fig. 2d, 4j**). We revised the Discussion section accordingly.

- Line 121: *“PARP10 is required for suppressing the accumulation of ssDNA gaps in wildtype cells” Both HeLa and DLD1 cells are cancer-derived cell lines, thus no WT cells. Either change the title or perform experiments in non-cancerous cell lines.*

We apologize for the confusion. By “wildtype” we meant “BRCA-proficient cells”. We revised the text accordingly (page 5). We believe this is an important issue to point out, as most previous studies on ssDNA gap accumulation employed BRCA-deficient cells.

- Line 201 and 202: *PLA demonstrates proximity, not interaction. Rephrase as in discussion.*

As mentioned above, in the revised manuscript, in addition to PLA assays, we present co-immunoprecipitation experiments to support the claim that PARP10 interacts with RAD18 (**new Fig. 2d, 4j**).

- Figure 2D: *PARP10KO condition was treated with 4mM HU instead of 0.4 mM HU. Is it a typo?*

We apologize for this. We had employed the wrong HU concentration in the PARP10^{KO} control sample. We now include a proper control using the correct concentration (0.4mM HU) (**new Supplementary Fig. S3b**).

- Line 207 – 209: *it is unclear how this data suggests a feedback loop. I would like a further explanation.*

For increased clarity, we changed the text to: “This potentially reflects the accumulation of ssDNA gaps in RAD18-deficient cells, caused by defective PCNA ubiquitination-mediated gap filling.” (page 9).

- Figure 3A: *ubPCNA SIRF experiments are missing NT conditions.*

In the revised manuscript, we now present the results of a ub-PCNA SIF experiment which includes the NT conditions (**new Supplementary Fig. S3c**).

- Line 234 – 235 and Figure 3H: missing stats comparing WT and RAD18 KO, as presented in Figure 3G.

We added the statistical comparison indicated by the reviewer to the figure, and included the statistical analysis in the Source Data file.

- Line 235: HeLa cells are not WT cells, correct accordingly.

We corrected the text to indicate that we were referring to wildtype HeLa cells, as opposed to the HeLa RAD18-knockout cells (page 10).

- Line 352 – 354: *“Indeed, in our previous studies we found that the impact of PARP10 loss on overall PCNA ubiquitination levels is less severe than that of RAD18 depletion” In figure 3A it is not evident. Please provide stats to support that claim in this figure.*

We corrected the sentence to indicate that the statement referred to our previous publications, in which we used western blot-based detection of PCNA ubiquitination (page 18). While the difference is less evident in the ^{Ub}PCNA SIF experiments presented in this manuscript, this trend is still observable. Beyond the difference in the readouts of the assays, also differences in siRNA efficiency between the experiments may play a role.

REVIEWER COMMENTS

Reviewer #1 (Remarks to the Author):

I am happy with the revised version of the manuscript, which I believe has improved his value by providing mechanistic insights and further controls.

Reviewer #2 (Remarks to the Author):

Khatib et al

The revision by Khatib and colleagues, in this reviewers view, is unsuitable for publication in Nature Communications due to the following primary concerns: 1, findings represent a limited advance to the field. 2, lack of evidence to indicate that gaps are a source of toxicity, as suggested in the abstract and in conclusions. 3, lack of biological replicates throughout.

It is unclear how it is possible to generate, rigorously, 32 new figure panels between 4 authors in the revision time period. Although perhaps the single biological replicate is the reason why this was possible. If there are two repeats, these should be included in the panels. At least 2 and preferably 3 biological replicates should be performed.

In rebuttal, it is stated MAR antibody didn't work for western. But then it is used for PLA. Suggesting possible non-specific reaction between antibodies.

The relationship between gaps and cellular toxicity is controversial. However, thankfully, a recent study by the Jasin group firmly establish gaps are not a source of toxicity as there are many pathways of fill in. Therefore, conclusions made here, that the role of parp10 in gap repair is the source of toxicity is not supported by evidence. The authors state parp10 has many cellular functions, including cell cycle, nfkb, caspase apoptosis, etc. They are not able to conclude that gaps, as opposed to other parp10 functions, are the source of toxicity.

Reviewer #3 (Remarks to the Author):

Revision Khatib et al.,:

The authors have performed an excellent job addressing all the major and minor points raised. Importantly, the addition of the co-Immunoprecipitation and in vitro MARYlation experiments further strengthens the link between RAD18 and PARP10.

However, some additional work needs be put in these new experiments before publication:

- While it is clear from Figure 2d and 5a that PARP10 interacts and MARYlates RAD18, the authors also conclude a loss of interaction of PARP10 Δ PIP and RAD18 (Lines 288-289). The experiment shown in Figure 4j does currently not support the conclusion. The input PARP10 Δ PIP is considerably lower compared to PARP10WT, which could explain the loss of PARP10 Δ PIP in the coIP. Are those variations in the input simply of technical nature. Then I would ask the authors to improve the overall technical quality of the experiment. It is also clear that this effect is not "black-and-white" and therefore a quantification of biological replicates of this experiment will be necessary as well.

Response to referees

Reviewer #1

I am happy with the revised version of the manuscript, which I believe has improved his value by providing mechanistic insights and further controls.

We thank the reviewer for their comments and for their helpful suggestions during the manuscript revision.

Reviewer #2

The revision by Khatib and colleagues, in this reviewers view, is unsuitable for publication in Nature Communications due to the following primary concerns: 1, findings represent a limited advance to the field. 2, lack of evidence to indicate that gaps are a source of toxicity, as suggested in the abstract and in conclusions. 3, lack of biological replicates throughout.

It is unclear how it is possible to generate, rigorously, 32 new figure panels between 4 authors in the revision time period. Although perhaps the single biological replicate is the reason why this was possible. If there are two repeats, these should be included in the panels. At least 2 and preferably 3 biological replicates should be performed.

The 32 new figure panels included:

- Survival experiments which were presented in replicates -e.g. Figs S8 and S9 (Figs. S10 and S11 in the re-revised manuscript);
- Control experiments (eg no drug treatment) requested by the reviewers, or replicates using slightly different treatment conditions to match other experiments shown in the original submission, as also requested by the original reviewers -e.g. Fig. S3, Fig. S5 (Fig. S6 in the re-revised manuscript);
- Conceptual replicates in similar setups (e.g. same results obtained with HU and cisplatin in the same cell line, or with the same drug treatment in different cell lines, or with knockdown of two different genes which have the same function) -e.g. Fig. S2, Fig. S6 (Fig.S8 in the re-revised manuscript);
- Control western blot experiments showing gene knockdowns.

In this new resubmission, we include replicates for the experiments for which replicates were not included in the previous revision. These include:

- SIRF with the PARP10 Δ UIM mutant (new Supplementary Fig. S4c,d);
- RAD18-PARP10 co-immunoprecipitation (new Supplementary Fig. S5a);
- PLA with RAD18 and PARP10 mutants (new Supplementary Fig. S5c);
- In vitro* MARylation of RAD18 by PARP10 (new Supplementary Fig. S7a);
- PLA with RAD18 and MAR (new Supplementary Fig. S7b);
- BrdU alkaline comet with FEN1 depletion (new Supplementary Fig. S9b);
- Neutral comet with PARP10 inhibitors (new Supplementary Fig. S10a).

In rebuttal, it is stated MAR antibody didn't work for western. But then it is used for PLA. Suggesting possible non-specific reaction between antibodies.

The fact that an antibody does not work in western blot, does not mean that is non-functional in other applications. We had previously shown that the MAR single antibody control does not show PLA foci, indicating its specificity in this assay (Fig. 5d). We now show in this revised figure panel that also the RAD18 single antibody control does not show foci. This indicates that there is no non-specific reaction in this assay.

The relationship between gaps and cellular toxicity is controversial. However, thankfully, a recent study by the Jasin group firmly establish gaps are not a source of toxicity as there are many pathways of fill in. Therefore, conclusions made here, that the role of parp10 in gap repair is the source of toxicity is not supported by evidence. The authors state parp10 has many cellular functions, including cell cycle, nfkb, caspase apoptosis, etc. They are not able to conclude that gaps, as opposed to other parp10 functions, are the source of toxicity.

We would like to respectfully point out that our work is not in conflict with the recent Jasin paper. In fact, we had already addressed this issue in the original submission, in the Discussion section (page 18): *“Recent studies using separation of function BRCA2 mutants had however suggested that BRCA2 promotes chemotherapy resistance primarily through HR, rather than gap suppression^{65, 66}. PARP10-knockout cells are not hypersensitive to cisplatin, even though they accumulate gaps under those conditions. This suggests that, as long as cells are able to repair the gaps through BRCA-mediated recombination, their accumulation is not cytotoxic.”*

As we discussed on page 19, the PARP10 inhibition phenotype mimics the previously described phenotype of REV1 inhibition (ref. 7). Since we show here that loss of PARP10 reduces REV1 function at gaps, these findings argue that it is indeed gap filling which causes the hypersensitivity. Nevertheless, we agree that it cannot be ruled out that other PARP10 functions contribute to this, and we acknowledge this possibility in the re-revised manuscript's Discussion section (page 19).

Reviewer #3

The authors have performed an excellent job addressing all the major and minor points raised. Importantly, the addition of the co-Immunoprecipitation and in vitro MARYlation experiments further strengthens the link between RAD18 and PARP10.

We thank the reviewer for their comments and for their helpful suggestions during the manuscript revision. We are glad that the reviewer found that we did *“an excellent job addressing all the points raised”*.

However, some additional work needs be put in these new experiments before publication:

- *While it is clear from Figure 2d and 5a that PARP10 interacts and MARYlates RAD18, the authors also conclude a loss of interaction of PARP10 Δ PIP and RAD18 (Lines 288-289). The experiment shown in Figure 4j does currently not support the conclusion. The input PARP10 Δ PIP is considerably lower compared to PARP10WT, which could explain the loss of PARP10 Δ PIP in the colP. Are those variations in the input simply of technical nature. Then I would ask the authors to improve the overall technical quality of the experiment. It is also clear*

that this effect is not “black-and-white” and therefore a quantification of biological replicates of this experiment will be necessary as well.

We agree with the reviewer that it is difficult to judge differences in the PARP10-RAD18 interaction from co-immunoprecipitation experiments. As described in our manuscript, this is exactly why we went on to employ quantifiable PARP10-RAD18 PLA experiments to investigate this. In any case, in the re-revised manuscript, we provide a replicate co-immunoprecipitation experiment (new Supplementary Fig. S5a), as well as a quantification of the amount of PARP10 co-precipitated with RAD18 from the two independent replicates (new Supplementary Fig. S5b), which do indicate a reduction in the interaction of RAD18 with the PARP10 Δ PIP mutant.

REVIEWERS' COMMENTS

Reviewer #3 (Remarks to the Author):

In the second revision of the paper "The mono-ADP-ribosyltransferase PARP10 promotes the repair of replication-associated nascent strand DNA gaps through RAD18-mediated Translesion Synthesis" the authors have addressed the remaining point of the revisions. We congratulate the authors on a high-quality paper.

Response to referees

Reviewer #3 (Remarks to the Author):

In the second revision of the paper “The mono-ADP-ribosyltransferase PARP10 promotes the repair of replication-associated nascent strand DNA gaps through RAD18-mediated Translesion Synthesis” the authors have addressed the remaining point of the revisions. We congratulate the authors on a high-quality paper.

We thank the reviewer for their comments and for their helpful suggestions during the manuscript revision.